# Approximating any Function via Coreset for Radial Basis Functions: Towards Provable Data Subset Selection For Efficient Neural Networks training

## Abstract

Radial basis function neural networks (*RBFNN*) are well-known for their capability to approximate any continuous function on a closed bounded set with arbitrary precision given enough hidden neurons. Coreset is a small weighted subset of an input set of items, that provably approximates their loss function for a given set of queries (models, classifiers, etc.). In this paper, we suggest the first coreset construction algorithm for *RBFNNs*, i.e., a small weighted subset which approximates the loss of the input data on any radial basis function network and thus approximates any function defined by an *RBFNN* on the big input data. This is done by constructing coresets for radial basis and Laplacian loss functions. We use our coreset to suggest a provable data subset selection algorithm for training deep neural networks, since our coreset approximates every function, it should approximate the gradient of each weight in a neural network as it is defined as a function on the input. Experimental results on function approximation and dataset subset selection on popular network architectures and data sets are presented, demonstrating the efficacy and accuracy of our coreset construction.

## 1 Introduction

Radial basis function neural networks (*RBFNNs*) are artificial neural networks that generally have three layers: an input layer, a hidden layer with a radial basis function (RBF) as an activation function, and a linear output layer. In this paper, the input layer receives a $d$-dimensional vector $x \in \mathbb{R}^d$ of real numbers. The hidden layer then consists of various nodes representing *RBFs*, to compute $\rho(\|x - c_i\|_2) := \exp\left(-\|x - c_i\|_2^2\right)$, where $c_i \in \mathbb{R}^d$ is the center vector for neuron $i$ across, say, $N$ neurons in the hidden layer. The linear output layer then computes $\sum_{i=1}^{N} \alpha_i \rho(\|x - c_i\|_2)$, where $\alpha_i$ is the weight of neuron $i$ in the linear output neuron. Therefore, *RBFNNs* are feed-forward neural networks because the edges between the nodes do not form a cycle, and enjoy advantages such as simplicity of analysis, faster training time, and interpretability, compared to alternatives such as convolutional neural networks (*CNNs*) and even multi-layer perceptrons (*MLPs*) (Padmavati, 2011).

**Function approximation via *RBFNNs*.** *RBFNNs* are universal approximators in the sense that an *RBFNN* with a sufficient number of hidden neurons (large $N$) can approximate any continuous function on a closed, bounded subset of $\mathbb{R}^d$ with arbitrary precision (Park & Sandberg, 1991), i.e., given a sufficiently large input set $P$ of $n$ points in $\mathbb{R}^d$ and given its corresponding label function $y : P \to \mathbb{R}$, an *RBFNN*, can be trained to approximate the function $y$. Therefore, *RBFNNs* are commonly used across a wide range of applications, such as function approximation (Park & Sandberg, 1991; 1993; Lu et al., 1997), time series prediction (Whitehead & Choate, 1996; Leung et al., 2001; Harpham & Dawson, 2006), classification (Leonard & Kramer, 1991; Wuxing et al., 2004; Babu & Suresh, 2012), and system control (Yu et al., 2011; Liu, 2013), due to their faster learning speed.

For a given size of RBFNN (number of neurons in the hidden layer) and an input set, the aim of this paper is to compute a small weighted subset that approximates the loss of the input data on any radial basis function neural network of this size and thus approximates any function defined

(approximated) by such an *RBFNN* on the big input data. This small weighted subset is denoted by coreset.

**Coresets.** Usually, in machine/deep learning, we are given input set $P \subseteq \mathbb{R}^d$ of $n$ points, its corresponding weights function $w : P \to \mathbb{R}$, a set of queries $X$ (a set of candidate solutions for the involved optimization problem), and a loss function $f : P \times X \to [0, \infty)$. The tuple $(P, w, X, f)$ is called *query space*, and it defines the optimization problem at hand — where usually, the goal is to find $x^* \in \arg\min_{x \in X} \sum_{p \in P} w(p) f(p, x)$. Given a query space $(P, w, X, f)$, a coreset for it is a small weighted subset of the input $P$ that can provably approximate the cost of every query $x \in X$ on $P$ (Feldman, 2020); see Definition 1. In particular, a coreset for a *RBFNN* can approximate the cost of an *RBFNN* on the original training data for every set of centers and weights that define the *RBFNN* (see Section 4). Hence, the coreset approximates also the centers and weights that form the optimal solution of the *RBFNN* (the solution that approximates the desired function). Thus a coreset for a *RBFNN* would facilitate training data for function approximation without reading the full training data and more generally, a strong coreset for an RBFNN with enough hidden neurons would give a strong coreset for any function that can be approximated to some precision using the RBFNN.

**To this end, in this paper, we aim to provide a coreset for *RBFNNs*, and thus provably approximating (providing a coreset to) any function that can be approximated by a given *RBFNN*.**

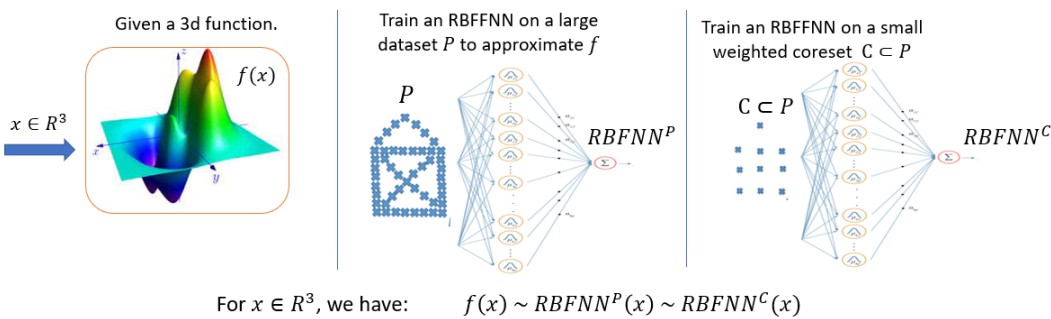

For $x \in R^3$, we have: $f(x) \sim RBFNN^P(x) \sim RBFNN^C(x)$

Figure 1: Our contribution in a nutshell.

Furthermore, we can use this small weighted subset (coreset) to suggest a provable data subset selection algorithm for training deep neural networks efficiently (on the small subset), since our coreset approximates every function that can be approximated by an RBFNN of this size, it should approximate the gradient of each weight in a neural network (if it can be approximated by the RBFNN).

**Training neural networks on data subset.** Although deep learning has become widely successful with the increasing availability of data (Krizhevsky et al., 2017; Devlin et al., 2019), modern deep learning systems have correspondingly increased in their computational resources, resulting in significantly larger training times, financial costs (Sharir et al., 2020), energy costs (Strubell et al., 2019), and carbon footprints (Strubell et al., 2019; Schwartz et al., 2020). Data subset selection (coresets) allows for efficient learning at several levels (Wei et al., 2014; Kaushal et al., 2019; Coleman et al., 2019; Har-Peled & Mazumdar, 2004; Clarkson, 2010). By employing a significantly smaller subset of the big dataset, (i) we enable learning on relatively low resource computing settings without requiring a huge number of GPU and CPU servers, (ii) we may greatly optimize the end-to-end turnaround time, which frequently necessitates many training runs for hyper-parameter tweaking, and (iii) because a large number of deep learning trials must be done in practice, we allow for considerable reductions in deep learning energy usage and $CO_2$ emissions (Strubell et al., 2019). Several efforts have recently been made to improve the efficiency of machine learning models using data subset selection (Mirzasoleiman et al., 2020a; Killamsetty et al., 2021b;a). However, existing techniques either (i) employ proxy functions to choose data points, (ii) are specialized to specific machine learning models, (iii) use approximations of parameters such as gradient error or generalization errors, (iv) lack provable guarantees on the approximation error, or (v) require an inefficient gradient computation of the whole data. Most importantly, all of these methods are model/network

dependent, and thus computing the desired subset of the data after several training epochs (for the same network) takes a lot of time and must be repeated each time the network changes.

**To this end, in this paper, we aim at suggesting a provable and efficient model-independent subset selection algorithm for training neural networks. This will allow us to compute a subset of the training data, that is guaranteed to be a coreset for training several neural network architectures/models.**

## 1.1 OUR CONTRIBUTIONS

In this paper, we suggest a coreset that approximates any function can be represented by an *RBFNN* architecture. Specifically:

(i) We provide a coreset for the *RBF* and Laplacian cost functions; see Algorithm 1, and Section 3.1.2.

(ii) We employ (i) to generate a coreset for any *RBFNN* model, in turn, approximating any function that can represented by the *RBFNN*; see Figure 1 for illustration, and Section 4 for more details.

(iii) We then exploit the properties of *RBFNNs*, to approximate the gradients of any deep neural networks (*DNNs*), leading towards provable subset selection for learning/training *DNNs*. We also show the advantages of using our coreset against previous subset selection techniques; see Section 5 and Section 6.

(iv) Finally, we provide an open source code implementation of our algorithm for reproducing our results and future research.

## 1.2 RELATED WORK

A long line of active work has studied efficient coreset constructions for various problems in computational geometry and machine learning, such as $k$-means and $k$-median clustering (Har-Peled & Mazumdar, 2004; Chen, 2009; Braverman et al., 2016; Huang & Vishnoi, 2020; Jubran et al., 2020; Cohen-Addad et al., 2022), regression (Dasgupta et al., 2008; Chhaya et al., 2020; Tolochinsky et al., 2022; Meyer et al., 2022), low-rank approximation (Cohen et al., 2017; Braverman et al., 2020; Maalouf et al., 2020; 2021), volume maximization (Indyk et al., 2020; Mahabadi et al., 2020; Woodruff & Yasuda, 2022), projective clustering (Feldman et al., 2020; Tukan et al., 2022b), support vector machines (SVMs) (Clarkson, 2010; Tukan et al., 2021), Bayesian inference (Campbell & Broderick, 2018), sine wave fitting (Maalouf et al., 2022), and decision tree (Jubran et al., 2021). (Baykal et al., 2022) suggested coreset-based algorithms for compressing the parameters of a trained fully-connected neural network by using sensitivity sampling on the weights of neurons after training, though without pruning full neurons. (Mussay et al., 2020; Liebenwein et al., 2019; Tukan et al., 2022a) sidestepped this issue by identifying the neurons that can be compressed regardless of their weights, due to the choice of the activation functions, thereby achieving coreset-based algorithms for neural pruning.

These approaches use coresets to achieve an orthogonal goal to data subset selection in the context of deep learning – they greatly reduce the number of neurons in the network while we greatly reduce the number of samples in the dataset that need to be read by the neural network. Correspondingly, we reduce the effective size of the data that needs to be stored or even measured prior to the training stage. Moreover, we remark that even if the number of inputs to the input layer were greatly reduced by these neural compression approaches, the union of the inputs can still consist of the entire input dataset and so these approaches generally cannot guarantee any form of data distillation.

Toward the goal of data subset selection, (Mirzasoleiman et al., 2020a;b) introduced algorithms for selecting representative subsets of the training data to accurately estimate the full gradient for tasks in both deep learning and classical machine learning models such as logistic regression and these approaches were subsequently refined by (Killamsetty et al., 2021a;b). Data distillation has also received a lot of attention in image classification (Bohdal et al., 2020; Nguyen et al., 2021; Dosovitskiy et al., 2021), natural language processing (Devlin et al., 2019; Brown et al., 2020), and federated learning (Ozkara et al., 2021; Zhu et al., 2021).

## 2 PRELIMINARIES

For an integer $n > 0$, we use $[n]$ to denote the set $\{1, 2, \ldots, n\}$. A weighted set of points is a pair $(P, w)$, where $P \subseteq \mathbb{R}^d$ is a set of points and $w : P \to [0, \infty)$ is a weight function.

We now formally provide the notion of $\varepsilon$-coreset for the *RBF* loss. This will be later extended to a coreset for *RBFNN*.

**Definition 1** (*RBF $\varepsilon$-coreset*). *Let $(P, w)$ be a weighted of $n$ points in $\mathbb{R}^d$, $X \subseteq \mathbb{R}^d$ be a set of queries, $\varepsilon \in (0, 1)$. For every $x \in X$ and $p \in P$ let $f(p, x) := \exp\left(-\|p - x\|_2^2\right)$ denote the* RBF *loss function between $p$ and $x$. An $\varepsilon$-coreset for $(P, w)$ with respect to $f$, is a pair $(S, v)$ where $S \subseteq P$, $v : S \to (0, \infty)$ is a weight function, such that for every $x \in X$, $\left|1 - \frac{\sum_{q \in S} v(q) f(q, x)}{\sum_{p \in P} w(p) f(p, x)}\right| \leq \varepsilon$.*

We say the *RBF* coreset is *strong* if it guarantees correctness over all $x \in X$. Otherwise, we say the coreset is *weak* if it only provides guarantees for all $x$ only in some subset of $X$.

**Sensitivity sampling.** To compute our *RBF $\varepsilon$-coreset*, we utilize the sensitivity sampling framework (Braverman et al., 2016). In short, the sensitivity of a point $p \in P$ corresponds to the "importance" of this point with respect to the other points and the problem at hand. In our context (with respect to the *RBF* loss), the sensitivity is defined as $s(p) = \sup_{x \in X} \frac{w(p) f(p, x)}{\sum_{q \in P} w(q) f(q, x)}$, where the denominator is nonzero. Once we bound the sensitivities for every $p \in P$, we can sample points from $P$ according to their corresponding sensitivity bounds, and re-weight the sampled points to obtain an RBF $\varepsilon$-coreset as in Definition 1. The size of the sample (coreset) is proportional to the sum of these bounds – the tighter (smaller) these bounds, the smaller the coreset size. For formal details, we refer the reader to Section A in the appendix.

**Sensitivity bounding.** We now present our main tool for bounding the sensitivity of each input point with respect to the *RBF* and *Laplacian* loss functions.

**Definition 2** (Special case of Definition 4 (Tukan et al., 2020)). *Let $\left(P, w, \mathbb{R}^d, f\right)$ be query space as in Definition 9 where for every $p \in P$ and $x \in \mathbb{R}^d$, $f(p, x) = \left|p^T x\right|$. Let $D \in [0, \infty)^{d \times d}$ be a diagonal matrix of full rank and let $V \in \mathbb{R}^{d \times d}$ be an orthogonal matrix, such that for every $x \in \mathbb{R}^d$,*

$$\left\|DV^T x\right\|_2 \leq \sum_{p \in P} w(p) \left|p^T x\right| \leq \sqrt{d} \left\|DV^T x\right\|_2.$$

*Define $U : P \to \mathbb{R}^d$ such that $U(p) = p\left(DV^T\right)^{-1}$ for every $p \in P$. The tuple $(U, D, V)$ is the $\|\cdot\|_1$-SVD of $P$.*

Using the above tool, the sensitivity w.r.t. the RBF loss function can be bounded using the following.

**Lemma 3** (Special case of Lemma 35, (Tukan et al., 2020)). *Let $\left(P, w, \mathbb{R}^d, f\right)$ be query space as in Definition 9 where for every $p \in P$ and $x \in \mathbb{R}^d$, $f(p, x) = \left|p^T x\right|$. Let $(U, D, V)$ be the $\|\cdot\|_1$-SVD of $(P, w)$ with respect to $|\cdot|$ (see Definition 2). Then, claims (i) – (ii) hold as follows:*

*(i) for every $p \in P$, the sensitivity of $p$ with respect to the query space $(P, w, \mathbb{R}^d, |\cdot|)$ is bounded by $s(p) \leq \|U(p)\|_1$,*

*(ii) and the total sensitivity is bounded by $\sum_{p \in P} s(p) \leq d^{1.5}$.*

## 3 METHOD

In this section, we provide coresets for the Gaussian and Laplacian loss functions. We detail our coreset construction for the Gaussian loss function and Laplacian loss function in Section 3.1.2.

**Overview of Algorithm 1.** Algorithm 1 receives as input, a set $P$ of $n$ points in $\mathbb{R}^d$, a weight function $w : P \to [0, \infty)$, a bound $R$ on the radius of the ball containing query space $X$, and a sample size $m > 0$. If the sample size $m$ is sufficiently large, then Algorithm 1 outputs a pair $(S, v)$ that is an $\varepsilon$-coreset for RBF cost function; see Theorem 6.

First $d'$ is set to be the VC dimension of the the quadruple $(P, w, X, \rho(\cdot))$; see Definition 10. The heart of our algorithm lies in formalizing the RBF loss function as a variant of regression problem, specifically, a variant of the $\ell_1$-regression problem. The conversion requires manipulation of the input data as presented at Line 2. We then compute the $f$-SVD of the new input data with respect to the $\ell_1$-regression problem followed by bounded the sensitivity of such points (Lines 3–5). Now we have all the needed ingredients to use Theorem 11 in order to obtain an $\varepsilon$-coreset, i.e., we sample i.i.d $m$ points from P based on their sensitivity bounds (see Line 8), followed by assigning a new weight for every sampled point at Line 9.

---

**Algorithm 1:** CORESET$(P, w, R, m)$

**Input:** A set $P \subseteq \mathbb{R}^d$ of $n$ points, a weight function $w : P \to [0, \infty)$, a bound on radius of the query space $X$, and a sample size $m \geq 1$.

**Output:** A pair $(S, v)$ that satisfies Theorem 6.

1 Set $d' :=$ the VC dimension of quadruple $(P, w, X, \rho(\cdot))$ // See Definition 10
2 Set $P' = \left\{ q_p = \left[ \|p\|_2^2, -2p^T, 1 \right]^T \mid \forall p \in P \right\}$
3 Set $(U, D, V)$ to be the $f$-SVD of $(P', w, |\cdot|)$ // See Definition 2
4 **for** *every $p \in P$* **do**
5 $\quad$ Set $s(p) := e^{12R^2} \left( 1 + 8R^2 \right) \left( \frac{w(p)}{\sum\limits_{q \in P} w(q)} + w(p) \|U(q_p)\|_1 \right)$
$\quad$ // bound on the sensitivity of $p$ as in Lemma 13 in the appendix
6 Set $t := \sum_{p \in P} s(p)$
7 Set $\tilde{c} \geq 1$ to be a sufficiently large constant // Can be determined from Theorem 6
8 Pick an i.i.d sample $S$ of $m$ points from $P$, where each $p \in P$ is sampled with probability $\frac{s(p)}{t}$.
9 set $v : \mathbb{R}^d \to [0, \infty]$ to be a weight function such that for every $q \in S$, $v(q) = \frac{t}{s(q) \cdot m}$.
10 **return** $(S, v)$

---

### 3.1 ANALYSIS

#### 3.1.1 LOWER BOUND ON THE CORESET SIZE FOR THE GAUSSIAN LOSS FUNCTION

We first show that the lower bound on the size of coresets, to emphasize the need for assumptions on the data and the query space.

**Theorem 4.** *There exists a set of $n$ points $P \subseteq \mathbb{R}^d$ such that $\sum_{p \in P} s(p) = \Omega(n)$.*

*Proof.* Let $d \geq 3$ and let $P \subseteq \mathbb{R}^d$ be a set of $n$ points distributed evenly on a 2 dimensional sphere of radius $\sqrt{\frac{\ln n}{2 \cos \left( \frac{\pi}{n} \right)}}$. In other words, using the law of cosines, every $p \in P$, $\sqrt{\ln n} = \min_{q \in P \setminus \{p\}} \|p - q\|_2$; see Figure C. Observe that for every $p \in P$,

$$s(p) := \max_{x \in \mathbb{R}^d} \frac{e^{-\|p-x\|_2^2}}{\sum\limits_{q \in P} e^{-\|q-x\|_2^2}} \geq \frac{e^{-\|p-p\|_2^2}}{\sum\limits_{q \in P} e^{-\|p-q\|_2^2}} = \frac{1}{1 + \sum\limits_{q \in P \setminus \{p\}} e^{-\|p-q\|_2^2}} \geq \frac{1}{1 + \sum\limits_{q \in P \setminus \{p\}} \frac{1}{n}} \geq \frac{1}{2},$$

(1)

where the first equality holds by definition of the sensitivity, the first inequality and second equality hold trivially, the second inequality follows from the assumption that $\sqrt{\ln n} \leq \min_{q \in P \setminus \{p\}} \|p - q\|_2$, and finally the last inequality holds since $\sum_{q \in P \setminus \{p\}} \frac{1}{n} \leq 1$. $\quad \square$

#### 3.1.2 REASONABLE ASSUMPTIONS LEAD TO EXISTENCE OF CORESETS

Unfortunately, it is not immediately straightforward to bound the sensitivities of either the Gaussian loss function or the Laplacian loss function. Therefore, we first require the following structural properties in order to relate the Gaussian and Laplacian loss functions into more manageable quantities.

We shall ultimately relate the function $e^{-|p^T x|}$ to both the Gaussian and Laplacian loss functions. Thus, we first relate the function $e^{-|p^T x|}$ to the function $|p^T x| + 1$.

**Claim 5.** *Let $p \in \mathbb{R}^d$ such that $\|p\|_2 \leq 1$, and let $R > 0$ be positive real number. Then for every $x \in \{x \in \mathbb{R}^d \,\big|\, \|x\|_2 \leq R\}$,*

$$\frac{1}{e^R (1+R)} \left(1 + |p^T x|\right) \leq e^{-|p^T x|} \leq |p^T x| + 1.$$

In what follows, we provide the analysis of coreset construction for the RBF and Laplacian loss functions, considering an input set of points lying in the unit ball. We refer the reader to the supplementary material for generalization of our approaches towards general input set of points.

**Theorem 6** (Coreset for *RBF*). *Let $R \geq 1$ be a positive real number, $X = \{x \in \mathbb{R}^d \,\big|\, \|x\|_2 \leq R\}$, and let $\varepsilon, \delta \in (0,1)$. Let $(P, w, X, f)$ be query space as in Definition 9 such that every $p \in P$ satisfies $\|p\|_2 \leq 1$. For every $x \in X$ and $p \in P$, let $f(p,x) := \rho\left(\|p - x\|_2\right)$. Let $(S, v)$ be a call to $\mathrm{CORESET}(P, w, R, m)$ where $S \subseteq P$ and $v : S \to [0, \infty)$. Then $(S, v)$ $\varepsilon$-coreset of $(P, w)$ with probability at least $1 - \delta$, if $m = O\left(\frac{e^{12R^2} R^2 d^{1.5}}{\varepsilon^2} \left(R^2 + \log d + \log \frac{1}{\delta}\right)\right)$.*

**Coreset for Laplacian loss function.** In what follows, we provide a coreset for the Laplacian loss function. Intuitively speaking, leveraging the properties of the Laplacian loss function, we were able to construct a coreset that holds for every vector $x \in \mathbb{R}^d$ unlike the *RBF* case where the coreset holds for a ball of radius $R$. We stress out that the reason for this is due to the fact that the Laplacian loss function is less sensitive than the *RBF*.

**Theorem 7** (Coreset for the Laplacian loss function). *Let $\left(P, w, \mathbb{R}^d, f\right)$ be query space as in Definition 9 such that every $p \in P$ satisfies $\|p\|_2 \leq 1$. For $x \in \mathbb{R}^d$ and $p \in P$, let $f(p,x) := e^{-\|p-x\|_2}$. Let $\varepsilon, \delta \in (0,1)$. Then there exists an algorithm which given $P, w, \varepsilon, \delta$ return a weighted set $(S, v)$ where $S \subseteq P$ of size $O\left(\frac{\sqrt{n} d^{1.25}}{\varepsilon^2} \left(\log n + \log d + \log \frac{1}{\delta}\right)\right)$ and a weight function $v : S \to [0, \infty)$ such that $(S, v)$ is an $\varepsilon$-coreset of $(P, w)$ with probability at least $1 - \delta$.*

## 4 RADIAL BASIS FUNCTION NETWORKS

In this section, we consider coresets for *RBFNNs*. Consider an *RBFNN* with $L$ neurons in the hidden layer and a single output neuron. First note that the hidden layer uses radial basis functions as activation functions so that the output is a scalar function of the input layer, $\phi : \mathbb{R}^d \to \mathbb{R}$ defined by $\phi(x) = \sum_{i=1}^{L} \alpha_i \rho(\|x - c^{(i)}\|_2)$, where $c^{(i)} \in \mathbb{R}^n$ for each $i \in [d]$.

For an input dataset $P$ and a corresponding desired output function $y : P \to \mathbb{R}$, RBFNNs aim to optimize

$$\sum_{p \in P} \left(y(p) - \sum_{i=1}^{L} \alpha_i e^{-\|p - c^{(i)}\|_2^2}\right)^2.$$

Expanding the above cost function, we obtain that RBFNNs aim to optimize

$$\sum_{p \in P} y(p)^2 - 2 \underbrace{\sum_{i=1}^{L} \alpha_i \left(\sum_{p \in P} y(p) e^{-\|p - c^{(i)}\|_2^2}\right)}_{\alpha} + \overbrace{\sum_{p \in P} \left(\sum_{i=1}^{L} \alpha_i e^{-\|p - c^{(i)}\|_2^2}\right)^2}^{\beta}. \tag{2}$$

**Bounding the $\alpha$ term in equation 2.** We first define for every $x \in \mathbb{R}^d$:

$$\phi^+(x) = \sum_{p \in P, y(p) > 0} y(p) e^{-\|p - x\|_2^2}$$

$$\phi^-(x) = \sum_{p \in P, y(p) < 0} |y(p)| \, e^{-\|p - x\|_2^2}.$$

Observe that $\sum_{p \in P} y(p)\rho(\|p - c^{(i)}\|_2) = \phi^+\left(c^{(i)}\right) - \phi^-\left(c^{(i)}\right)$. Thus the $\alpha$ term in equation 2 can be approximated using the following.

**Theorem 8.** *There exists an algorithm that samples* $O\left(\frac{e^{8R^2}R^2 d^{1.5}}{\varepsilon^2}\left(R^2 + \log d + \log \frac{2}{\delta}\right)\right)$ *points to form weighted sets* $(S_1, w_1)$ *and* $(S_2, w_2)$ *such that with probability at least* $1 - 2\delta$,

$$\frac{\left| \sum_{p \in P} y(p)\phi(p) - \left( \sum_{\substack{i \in [L] \\ \alpha_i > 0}} \alpha_i \sum_{p \in S_1} w_1(p) e^{-\|p - c^{(i)}\|_2^2} + \sum_{\substack{j \in [L] \\ \alpha_j < 0}} \alpha_j \sum_{q \in S_2} w_2(q) e^{-\|q - c^{(j)}\|_2^2} \right) \right|}{\left| \sum_{\substack{i \in [L] \\ \alpha_i > 0}} \alpha_i \left( \phi^+\left(c^{(i)}\right) + \phi^-\left(c^{(i)}\right) \right) \right| + \left| \sum_{\substack{i \in [L] \\ \alpha_i < 0}} |\alpha_i| \left( \phi^+\left(c^{(i)}\right) + \phi^-\left(c^{(i)}\right) \right) \right|} \leq \varepsilon.$$

**Bounding the $\beta$ term in equation 2.** By Cauchy's inequality, it holds that

$$\sum_{p \in P} \left( \sum_{i=1}^{L} \alpha_i e^{-\|p - c^{(i)}\|_2^2} \right)^2 \leq L \sum_{p \in P} \sum_{i=1}^{L} \alpha_i^2 e^{-2\|p - c^{(i)}\|_2^2} = L \sum_{i=1}^{L} \alpha_i^2 \sum_{p \in P} e^{-2\|p - c^{(i)}\|_2^2},$$

where the equality holds by simple rearrangement.

Using Theorem 6, we can approximate the upper bound on $\beta$ with an approximation of $L(1 + \varepsilon)$. However, if for every $i \in [L]$ it holds that $\alpha_i \geq 0$, then we also have the lower bound

$$\sum_{i=1}^{L} \alpha_i^2 \sum_{p \in P} e^{-2\|p - c^{(i)}\|_2^2} \leq \sum_{p \in P} \left( \sum_{i=1}^{L} \alpha_i e^{-\|p - c^{(i)}\|_2^2} \right)^2.$$

Since we can generate a multiplicative coreset for the left-hand side of the above inequality, then we obtain also a multiplicative coreset in a sense for $\beta$ as well.

## 5 BENEFIT OF OUR CORESET OVER OTHER SUBSET SELECTION METHODS

**One coreset for all networks.** Our coreset is a model independent, i.e., we aim at improving the running time of multiple neural networks. Contrary to other method that needs to compute the coreset after each gradient update to support there theoretical proofs, our method gives the advantage of computing the sensitivity (or the coreset) only once, for all of the required networks. This is since our coreset can approximate any function that can be defined (approximated) using a *RBFNN* model.

**Efficient coreset per epoch.** Practically, our competing methods for data selection are not applied before each epoch, but every $R$ epochs. This is since the competing methods require a lot of time to compute a new coreset since they compute the gradients of the network with respect to each input training data. However, our coreset can be computed before each epoch in a negligible time ($\sim 0$ seconds), since we compute the sensitivity of each point (image) in the data once at the beginning, and then whenever we need to create a new coreset, we simply sample from the input data according to the sensitivity distribution.

## 6 EXPERIMENTAL RESULTS

In this section we practically demonstrate the efficiency and stability of our *RBFNN* coreset approach for training deep neural networks via data subset selection. We mainly study the trade-off between accuracy and efficiency (time/subset size).

**Competing methods.** We compare our method against many variants of the proposed algorithms in Killamsetty et al. (2021a) (denoted by, *GRAD-MATCH*), in Mirzasoleiman et al. (2020a) (denoted by *CRAIG*), and in Killamsetty et al. (2021b) (denoted by *GLISTER*). For each of these methods, we report the results for 4 variants: (i) the "vanilla" method, denoted by its original name, (ii)

applying a warm start i.e., training on the whole data for $50\%$ of the training time before training the other $50\%$ on the coreset, where such methods are denoted by adding the suffix -WARM. (iii) a more efficient version of each of the competing methods denoted by adding the suffix PB (more details are given at Killamsetty et al. (2021a)), and finally, a combination of both (ii) and (iii). In other words, the competing methods are GRAD-MATCH, GRAD-MATCHPB, GRAD-MATCH-WARM, GRAD-MATCHPB-WARM, CRAIG, CRAIGPB, CRAIG-WARM, CRAIGPB-WARM, and GLISTER-WARM. We also compare against randomly selecting points (denoted by RANDOM).

**Datasets and model architecture.** We performed our experiments for training CIFAR10 and CIFAR100 (Krizhevsky et al., 2009) on Resnet18 (He et al., 2016), and MNIST (LeCun et al., 1998) on LeNet.

**The setting.** We adapted the same setting of Killamsetty et al. (2021a), where we used SGD optimizer for training initial learning rate equal to $0.01$, a momentum of $0.9$, and a weight decay of $5e - 4$. We decay the learning rate using cosine annealing (Loshchilov & Hutter, 2016) for each epoch. For MNIST, we trained the LeNet model for 200 epochs. For CIFAR10 and CIFAR100, we trained the ResNet18 for 300 epochs - all on batches of size 20 for the subset selection training versions. We train the data selection methods and the entire data training with the same number of epochs; the main difference is the number of samples used for training a single epoch. All experiments were executed on V100 GPUs. The reported test accuracy in the results is after averaging across five runs.

**Subset sizes and the $R$ parameter.** For MNIST, we use sizes of $\{1\%, 3\%, 5\%, 10\%\}$, while for CIFAR10 and CIFAR100, we use $\{5\%, 10\%, 20\%, 30\%\}$ on ResNet18. Since the competing methods require a lot of time to compute the gradients, we set $R = 20$. We note that for our coreset we can test it with $R = 1$ without adding run-time since once the sensitivity vector is defined, commuting a new coreset requires $\sim 0$ seconds. However, we test it with $R = 20$, to show its robustness.

**Discussion.** Table 1 and 2 report the results for CIFAR10 and CIFAR100 on ResNet18, respectively. It is clear from Table 1 that our method achieves the best accuracy (without warm start) for $20\%$ and $30\%$ subset selection of CIFAR10, while for CIFAR100 and smaller subset selection on CIFAR10, our method drastically outperform all of the methods that does not apply warm start (training on the whole data), and we still outperform all of the other methods in terms of accuracy vs time. The same phenomena is witnessed in the MNIST experiment (Table 4), where our coreset results are consistently placed among the top methods across all sizes. Furthermore, we note that our sensitivity sampling vector is computed once during our experiments for each dataset. This vector can be used to sampled coresets of different sizes, for different networks, at different epochs of training, in a time that is close to zero seconds. Note that the best results are represented in bold text, while the fastest are underlined.

Table 1: Data Selection Results for CIFAR10 using ResNet-18

| | Top-1 Test accuracy of the Model(%) | | | | Model Training time(in hrs) | | | |
|---|---|---|---|---|---|---|---|---|
| Budget(%) | 5% | 10% | 20% | 30% | 5% | 10% | 20% | 30% |
| FULL (skyline for test accuracy) | 95.09 | 95.09 | 95.09 | 95.09 | 4.34 | 4.34 | 4.34 | 4.34 |
| RANDOM (skyline for training time) | 71.2 | 80.8 | 86.98 | 87.6 | 0.22 | 0.46 | 0.92 | 1.38 |
| RANDOM-WARM (skyline for training time) | 83.2 | 87.8 | 90.9 | 92.6 | 0.21 | 0.42 | 0.915 | 1.376 |
| GLISTER | 85.5 | 91.92 | 92.78 | 93.63 | 0.43 | 0.91 | 1.13 | 1.46 |
| GLISTER-WARM | 86.57 | 91.56 | 92.98 | 94.09 | 0.42 | 0.88 | 1.08 | 1.40 |
| CRAIG | 82.74 | 87.49 | 90.79 | 92.53 | 0.81 | 1.08 | 1.45 | 2.399 |
| CRAIG-WARM | 84.48 | 89.28 | 92.01 | 92.82 | 0.6636 | 0.91 | 1.31 | 2.20 |
| CRAIGPB | 83.56 | 88.77 | 92.24 | 93.58 | 0.4466 | 0.70 | 1.13 | 2.07 |
| CRAIGPB-WARM | 86.28 | 90.07 | 93.06 | 93.8 | 0.4143 | 0.647 | 1.07 | 2.06 |
| GRADMATCH | 86.7 | 90.9 | 91.67 | 91.89 | 0.40 | 0.84 | 1.42 | 1.52 |
| GRADMATCH-WARM | 87.2 | 92.15 | 92.11 | 92.01 | 0.38 | 0.73 | 1.24 | 1.41 |
| GRADMATCHPB | 85.4 | 90.01 | 93.34 | 93.75 | 0.36 | 0.69 | 1.09 | 1.38 |
| GRADMATCHPB-WARM | 86.37 | **92.26** | 93.59 | 94.17 | 0.32 | 0.62 | 1.05 | 1.36 |
| RBFNN CORESET (OURS) | 86.9 | 91.4 | 93.61 | 94.44 | 0.28 | 0.52 | 0.98 | 1.38 |
| RBFNN CORESET-WARM (OURS) | **87.82** | 91.44 | **93.81** | **94.6** | 0.27 | 0.51 | 0.99 | 1.34 |

Table 2: Data Selection Results for CIFAR100 using ResNet-18

| | Top-1 Test accuracy of the Model(%) | | | | Model Training time(in hrs) | | | |
|---|---|---|---|---|---|---|---|---|
| Budget(%) | 5% | 10% | 20% | 30% | 5% | 10% | 20% | 30% |
| FULL (skyline for test accuracy) | 75.37 | 75.37 | 75.37 | 75.37 | 4.871 | 4.871 | 4.871 | 4.871 |
| RANDOM (skyline for training time) | 19.02 | 31.56 | 49.6 | 58.56 | 0.2475 | 0.4699 | 0.92 | 1.453 |
| RANDOM-WARM (skyline for training time) | 58.2 | 65.95 | 70.3 | 72.4 | 0.242 | 0.468 | 0.921 | 1.43 |
| GLISTER | 29.94 | 44.03 | 61.56 | 70.49 | 0.3536 | 0.6456 | 1.11 | 1.5255 |
| GLISTER-WARM | 57.17 | 64.95 | 62.14 | 72.43 | 0.3185 | 0.6059 | 1.06 | 1.452 |
| CRAIG | 36.61 | 55.19 | 66.24 | 70.01 | 1.354 | 1.785 | 1.91 | 2.654 |
| CRAIG-WARM | 57.44 | 67.3 | 69.76 | 72.77 | 1.09 | 1.48 | 1.81 | 2.4112 |
| CRAIGPB | 38.95 | 54.59 | 67.12 | 70.61 | 0.4489 | 0.6564 | 1.15 | 1.540 |
| CRAIGPB-WARM | 57.66 | 67.8 | 70.84 | 73.79 | 0.394 | 0.6030 | 1.10 | 1.5567 |
| GRADMATCH | 41.01 | 59.88 | 68.25 | 71.5 | 0.5143 | 0.8114 | 1.40 | 2.002 |
| GRADMATCH-WARM | 57.72 | 68.23 | 71.34 | 74.06 | 0.3788 | 0.7165 | 1.30 | 1.985 |
| GRADMATCHPB | 40.53 | 60.39 | 70.88 | 72.57 | 0.3797 | 0.6115 | 1.09 | 1.56 |
| GRADMATCHPB-WARM | 58.26 | **69.58** | **73.2** | 74.62 | 0.300 | 0.5744 | 1.01 | 1.5683 |
| RBFNN CORESET (OURS) | 54.17 | 64.59 | 71.17 | 73.58 | 0.346 | 0.5699 | 1.01 | 1.552 |
| RBFNN CORESET-WARM (OURS) | **59.22** | 67.8 | 72.79 | **75.04** | 0.352 | 0.5710 | 1.03 | 1.56 |

Table 3: Data Selection Results for ImageNet2012 using ResNet-18

| | Top-1 Test accuracy of the Model(%) | Model Training time(in hrs) |
|---|---|---|
| Budget(%) | 5% | 5% |
| FULL (skyline for test accuracy) | 70.36 | 276.28 |
| RANDOM (skyline for training time) | 21.124 | 14.12 |
| CRAIGPB | 44.28 | 22.24 |
| GRADMATCH | 47.24 | 18.24 |
| GRADMATCHPB | 45.15 | 16.12 |
| RBFNN CORESET (OURS) | **47.26** | 15.24 |

## 6.1 FUNCTION APPROXIMATIONS

In this section, we compare our coreset to uniform for approximating the linear regression loss function by running RBFNN on the coreset, where the results are averaged across 8 trials, while the shaded regions correspond to the median absolute deviation. It is clear from Figure 2 that our coreset outperforms uniform sampling. Furthermore learning on the coreset was ×30 faster than learning on the whole data. The data set is the 3D spatial networks (Dua et al., 2017).

## 7 CONCLUSION AND FUTURE WORK

In this paper, we have introduced a coreset that provably approximates any function that can be represented by a RBFNN architecture. Our coreset construction can be used to approximate the gradients of any deep neural networks (*DNNs*), leading towards provable subset selection for learning/training *DNNs*. We also empirically demonstrate the value of our work by showing significantly better performances over various datasets and model architectures. As the first work on using coresets for data subset selection with respect to *RBFNNs*, our results lead to a number of interesting possible future directions. It is natural to ask whether there exist smaller coreset

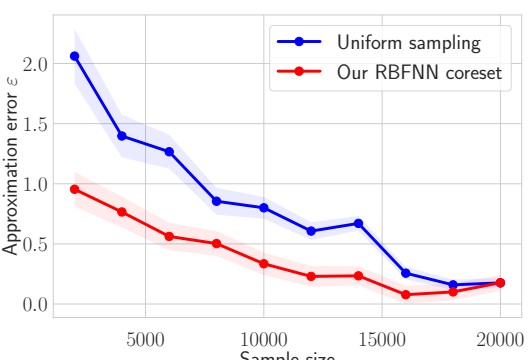

Figure 2: **Function approximation.**

constructions that also provably give the same worst-case approximation guarantees. Another question is whether our results can be extended to more general classes of loss functions. Finally, we remark that although our empirical results significantly beat state-of-the-art, they nevertheless only serve as a proof-of-concept and have not been fully optimized with additional heuristics.

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

## A  CORESET CONSTRUCTIONS

In what follows, we provide the necessary tools to obtain a coreset; see Definition 1.

**Definition 9** (Query space). *Let $P$ be a set of $n \geq 1$ points in $\mathbb{R}^d$, $w : P \to [0, \infty)$ be a non-negative weight function, and let $f : P \times \mathbb{R}^d \to [0, \infty)$ denote a loss function. The tuple $(P, w, \mathbb{R}^d, f)$ is called a query space.*

**Definition 10** (VC-dimension (Braverman et al., 2016)). *For a query space* $(P, w, \mathbb{R}^d, f)$ *and* $r \in [0, \infty)$, *we define*

$$\text{ranges}(x, r) = \{p \in P \mid w(p)f(p, x) \leq r\},$$

*for every* $x \in \mathbb{R}^d$ *and* $r \geq 0$. *The dimension of* $(P, w, \mathbb{R}^d, f)$ *is the size* $|S|$ *of the largest subset* $S \subset P$ *such that*

$$\left|\{S \cap \text{ranges}(x, r) \mid x \in \mathbb{R}^d, r \geq 0\}\right| = 2^{|S|},$$

*where* $|A|$ *denotes the number of points in $A$ for every* $A \subseteq \mathbb{R}^d$.

The following theorem formally describes how to construct an $\varepsilon$-coreset based on the sensitivity sampling framework.

**Theorem 11** (Restatement of Theorem 5.5 in (Braverman et al., 2016)). *Let* $(P, w, \mathbb{R}^d, f)$ *be a query space as in Definition 9. For every* $p \in P$ *define the* sensitivity *of $p$ as* $\sup_{x \in \mathbb{R}^d} \frac{w(p)f(p,x)}{\sum_{q \in P} w(q)f(q,x)}$, *where the sup is over every* $x \in \mathbb{R}^d$ *such that the denominator is non-zero. Let* $s : P \to [0, 1]$ *be a function such that $s(p)$ is an upper bound on the sensitivity of $p$. Let* $t = \sum_{p \in P} s(p)$ *and $d'$ be the* VC dimension *of the triplet* $(P, w, \mathbb{R}^d, f)$; *see Definition 10. Let* $c \geq 1$ *be a sufficiently large constant,* $\varepsilon, \delta \in (0, 1)$, *and let $S$ be a random sample of*

$$|S| \geq \frac{ct}{\varepsilon^2}\left(d' \log t + \log \frac{1}{\delta}\right)$$

*i.i.d points from $P$, such that every* $p \in P$ *is sampled with probability* $s(p)/t$. *Let* $v(p) = \frac{tw(p)}{s(p)|S|}$ *for every* $p \in S$. *Then, with probability at least* $1 - \delta$, $(S, v)$ *is an* $\varepsilon$-coreset *for $P$ with respect to $f$.*

# B PROOFS FOR OUR MAIN THEOREMS

## B.1 PROOF OF CLAIM 5

**Claim 5.** *Let* $p \in \mathbb{R}^d$ *such that* $\|p\|_2 \leq 1$, *and let* $R > 0$ *be positive real number. Then for every* $x \in \{x \in \mathbb{R}^d \mid \|x\|_2 \leq R\}$,

$$\frac{1}{e^R(1+R)}\left(1 + |p^T x|\right) \leq e^{-|p^T x|} \leq |p^T x| + 1.$$

*Proof.* Put $x \in \{x \in \mathbb{R}^d \mid \|x\|_2 \leq R\}$ and note that if $p^T x = 0$ then the claim is trivial. Otherwise, we observe that

$$e^{-|p^T x|} \geq \frac{1}{e^R} \geq \frac{1 + |p^T x|}{(1 + R)e^R}.$$

$\square$

## B.2 PROOF OF THEOREM 6

**Claim 12.** *Let* $a, b \geq 0$ *be pair of nonnegative real numbers and let* $c, d > 0$ *be a pair of positive real numbers. Then*

$$\frac{a+b}{c+d} \leq \frac{a}{c} + \frac{b}{d}.$$

*Proof.* Observe that

$$\frac{a+b}{c+d} = \frac{a}{c+d} + \frac{b}{c+d} \leq \frac{a}{c} + \frac{b}{d}$$

where the inequality holds since $c, d > 0$. $\square$

**Lemma 13** (Sensitivity bound w.r.t. the *RBF* loss function). *Let* $R \geq 1$ *be a positive real number, and let* $X = \{x \in \mathbb{R}^d \mid \|x\|_2 \leq R\}$. *Let* $(P, w, \mathbb{R}^d, f)$ *be query space as in Definition 9 where for every* $p \in P$ *and* $x \in \mathbb{R}^d$, $f(p, x) = e^{-\|p-x\|_2^2}$. *Let* $P' := \left\{q_p = \left[\|p\|_2^2, -2p^T, 1\right]^T \mid \forall p \in P\right\}$ *and*

*let* $q^* \in \arg\sup_{q \in P'} e^{3R^2 \|q\|_2} \left(1 + 3R^2 \|q\|_2\right)$. *Let* $u(p) := \frac{w(p)}{e^{3R^2 \|q_p\|_2} \left(1 + 3R^2 \|q_p\|_2\right)}$ *and let* $(U, D, V)$ *be the* $\|\cdot\|_1$*-SVD of* $(P', u(\cdot))$. *Then for every* $p \in P$,

$$s(p) \le e^{3R^2 \|q_p\|_2} \left(1 + 3R^2 \|q_p\|_2\right) \left( \frac{u(p)}{\sum_{p' \in P} u(p')} + u(p) \|U(q_p)\|_1 \right),$$

*and*

$$\sum_{q \in P} s(q) \le e^{3R^2 \|q^*\|_2} \left(1 + 3R^2 \|q^*\|_2\right) \left(1 + (d+2)^{1.5}\right).$$

*Proof.* Let $X = \left\{ x \in \mathbb{R}^d \big| \|x\|_2 \le 1 \right\}$, and observe that for every $p \in P$ and $x \in \mathbb{R}^d$, it holds that

$$\|p - x\|_2^2 = \left| q_p^T y \right|, \tag{3}$$

where $q_p = \left[ \|p\|_2^2, -2p^T, 1 \right]^T$ and $y = \left[ 1, x, \|x\|_2^2 \right]^T$.

Let $Y = \left\{ \left[ 1, x, \|x\|_2^2 \right]^T \big| x \in X \right\}$. Following the definition of $Y$, for every $y \in Y$, we obtain that $\|y\|_2 \le 3R^2$. Hence, by plugging $p := \frac{q_p}{\|q_p\|}$ and $R := 3R^2 \|q_p\|$ for every $p \in P$ into Claim 5, we obtain that the for every $y \in Y$ and $p \in P$,

$$\frac{1}{e^{3R^2 \|q_p\|_2} \left(1 + 3R^2 \|q_p\|_2\right)} \left(1 + \left| q_p^T y \right|\right) \le e^{-\left| q_p^T y \right|} \le 1 + \left| q_p^T y \right|. \tag{4}$$

Note that for every $p \in P$, $u(p) := \frac{w(p)}{e^{3 \|q_p\|_2} \left(1 + 3 \|q_p\|_2\right)}$. Thus

$$\sup_{x \in X} \frac{w(p) f(p, x)}{\sum_{q \in P} w(q) f(q, x)}$$

$$= \sup_{y \in Y} \frac{w(p) e^{-\left| q_p^T y \right|}}{\sum_{p' \in P'} w(q) \left| q_{p'}^T y \right|}$$

$$\le e^{3R^2 \|q_p\|_2} \left(1 + 3R^2 \|q_p\|_2\right) \sup_{y \in Y} \frac{u(p) \left| q_p^T y \right| + u(p)}{\sum_{p' \in P} u(p') \left| q_{p'}^T y \right| + \sum_{p' \in P} u(p')} \tag{5}$$

$$\le e^{3R^2 \|q_p\|_2} \left(1 + 3R^2 \|q_p\|_2\right) \left( \frac{u(p)}{\sum_{p' \in P} u(p')} + \sup_{y \in Y} \frac{u(p) \left| q_p^T y \right|}{\sum_{p' \in P} u(p') \left| q_{p'}^T y \right|} \right),$$

where the first inequality holds by Claim 5 and the second inequality is by Claim 12.

Let $f : P' \times Y \to [0, \infty)$ be a function such that for every $q \in P'$ and $y \in Y$, $f(q, y) = \left| q^T y \right|$. Plugging in $P := P'$, $w := u$, $d := d + 2$, and $f := f$ into Lemma 3 yields for every $p \in P$

$$\sup_{x \in X} \frac{w(p) f(p, x)}{\sum_{q \in P} w(q) f(q, x)} \le e^{3R^2 \|q_p\|_2} \left(1 + 3R^2 \|q_p\|_2\right) \left( \frac{u(p)}{\sum_{p' \in P} u(p')} + u(p) \|U(q_p)\|_1 \right). \tag{6}$$

Note that by definition, $q^* \in \arg\sup_{q \in P'} e^{3R^2 \|q\|_2} \left(1 + 3R^2 \|q\|_2\right)$. Then the total sensitivity is bounded by

$$\sum_{q \in P} \sup_{x \in X} \frac{w(p) f(p, x)}{\sum_{q \in P} w(q) f(q, x)} \le e^{3R^2 \|q^*\|_2} \left(1 + 3R^2 \|q^*\|_2\right) \left(1 + (d+2)^{1.5}\right). \tag{7}$$

$\square$

**Theorem 6** (Coreset for *RBF*). *Let $R \geq 1$ be a positive real number, $X = \{x \in \mathbb{R}^d \big| \|x\|_2 \leq R\}$, and let $\varepsilon, \delta \in (0, 1)$. Let $(P, w, X, f)$ be query space as in Definition 9 such that every $p \in P$ satisfies $\|p\|_2 \leq 1$. For every $x \in X$ and $p \in P$, let $f(p, x) := \rho(\|p - x\|_2)$. Let $(S, v)$ be a call to* CORESET$(P, w, R, m)$ *where $S \subseteq P$ and $v : S \to [0, \infty)$. Then $(S, v)$ $\varepsilon$-coreset of $(P, w)$ with probability at least $1 - \delta$, if $m = O\left(\frac{e^{12R^2} R^2 d^{1.5}}{\varepsilon^2}\left(R^2 + \log d + \log\frac{1}{\delta}\right)\right)$.*

*Proof.* First, by plugging in the query space $(P, w, \mathbb{R}^d, f)$ into Lemma 13, we obtain that a bound on the sensitivities $s(p)$ for every $p \in P$ and a bound on the total sensitivities $t := e^{12R^2}\left(1 + 12R^2\right)\left(1 + (d+2)^{1.5}\right)$, since the $\max_{q \in P} \|q\|_2 \leq 1$. Notice that the analysis done in Lemma 13 is analogues to the steps done in Algorithm 1.

By plugging the bounds on the sensitivities, the bound on the total sensitivity $t$, probability of failure $\delta \in (0, 1)$, and approximation error $\varepsilon \in (0, 1)$ into Theorem 11, we obtain a subset $S' \subseteq Q$ and $v' : S' \to [0, \infty)$ such that the tuple $(S', v')$ is an $\varepsilon$-coreset for $(P, w)$ with probability at least $1 - \delta$. $\qquad\square$

### B.3 PROOF OF THEOREM 7

**Lemma 14** (Sensitivity bound w.r.t. the *Laplacian* loss function). *Let $\left(P, w, \mathbb{R}^d, f\right)$ be query space as in Definition 9 where for every $p \in P$ and $x \in \mathbb{R}^d$, $f(p, x) = e^{-\|p - x\|_2}$. Let $P' := \left\{q_p = \left[\|p\|_2^2, -2p^T, 1\right]^T \mid \forall p \in P\right\}$ and let $q^* \in \arg\sup_{q \in P'} e^{3\sqrt{\|q\|_2}}\left(1 + 3\sqrt{\|q\|_2}\right)$. Let $u(p) := \frac{w(p)}{e^{3\sqrt{\|q_p\|_2}}\left(1 + 3\sqrt{\|q_p\|_2}\right)}$ and let $(U, D, V)$ be the $\|\cdot\|_1$-SVD of $\left(P', u^2(\cdot)\right)$. Then for every $p \in P$,*

$$s(p) \leq e^{3\sqrt{\|q_p\|_2}}\left(1 + 3\sqrt{\|q_p\|_2}\right)\left(\frac{u(p)}{\sum_{p' \in P} u(p')} + u(p)\sqrt{\|U(q_p)\|_1}\right) + \frac{e^{\|p\|_2 + \sqrt{\|q^*\|_2}} w(p)}{\sum_{q \in P} w(q)},$$

*and*

$$\sum_{q \in P} s(q) \leq 2e^{3\sqrt{\|q^*\|_2}} + e^{3\sqrt{\|q^*\|_2}}\left(1 + 3\sqrt{\|q^*\|_2}\right)\left(1 + \sqrt{n}\,(d+2)^{1.25}\right).$$

*Proof.* Let $X = \left\{x \in \mathbb{R}^d \big| \|x\|_2 \leq 1\right\}$, and observe that for every $p \in P$ and $x \in \mathbb{R}^d$, it holds that

$$\|p - x\|_2 = \sqrt{\left|q_p^T y\right|}, \tag{8}$$

where $q_p = \left[\|p\|_2^2, -2p^T, 1\right]^T$ and $y = \left[1, x, \|x\|_2^2\right]^T$.

Let $Y = \left\{\left[1, x, \|x\|_2^2\right]^T \big| x \in X\right\}$. Hence, following Theorem 11, the sensitivity of each point $p \in P$, can be rewritten as

$$\sup_{x \in \mathbb{R}^d} \frac{w(p)f(p, x)}{\sum_{q \in P} w(q)f(q, x)} \leq \sup_{x \in X} \frac{w(p)f(p, x)}{\sum_{q \in P} w(q)f(q, x)} + \sup_{x \in \mathbb{R}^d \setminus X} \frac{w(p)f(p, x)}{\sum_{q \in P} w(q)f(q, x)}. \tag{9}$$

From here, we bound the sensitivity with respect to subspaces of $\mathbb{R}^d$.

**Handling queries from $X$.** Following the definition of $Y$, for every $y \in Y$, we obtain that $\|y\|_2 \leq 3$. Hence, by plugging $p := q_p$ for every $p \in P$ and $R := 3\|q_p\|_2$ into Claim 5, we obtain that the for every $y \in Y$ and $p \in P$,

$$\frac{1}{e^{3\sqrt{\|q_p\|_2}}\left(1 + 3\sqrt{\|q_p\|_2}\right)}\left(1 + \sqrt{\left|q_p^T y\right|}\right) \leq e^{-\sqrt{\left|q_p^T y\right|}} \leq 1 + \sqrt{\left|q_p^T y\right|}. \tag{10}$$

Note that for every $p \in P$, $u(p) := \frac{w(p)}{e^3\sqrt{\|q_p\|_2}\left(1+3\sqrt{\|q_p\|_2}\right)}$. Combining equation 9 and equation 10, yields that

$$\sup_{x \in X} \frac{w(p)f(p,x)}{\sum_{q \in P} w(q)f(q,x)} \le e^3\sqrt{\|q_p\|_2}\left(1+3\sqrt{\|q_p\|_2}\right) \sup_{y \in Y} \frac{u(p)\sqrt{\left|q_p^T y\right|} + u(p)}{\sum_{p' \in P} u(p')\sqrt{\left|q_{p'}^T y\right|} + \sum_{p' \in P} w(p')}$$

$$\le e^3\sqrt{\|q_p\|_2}\left(1+3\sqrt{\|q_p\|_2}\right)\left(\frac{u(p)}{\sum_{p' \in P} u(p')} + \sup_{y \in Y} \frac{u(p)\sqrt{\left|q_p^T y\right|}}{\sum_{p' \in P} u(p')\sqrt{\left|q_{p'}^T y\right|}}\right), \tag{11}$$

where the first inequality holds by Claim 5 and the second inequality is by Claim 12.

By Cauchy-Schwartz inequality,

$$\sup_{y \in Y} \frac{u(p)\sqrt{\left|q_p^T y\right|}}{\sum_{p' \in P} u(p')\sqrt{\left|q_{p'}^T y\right|}} = \sup_{y \in Y} \frac{\sqrt{u(p)^2\left|q_p^T y\right|}}{\sum_{p' \in P} \sqrt{u(p')^2\left|q_{p'}^T y\right|}}$$

$$\le \sup_{y \in Y} \frac{\sqrt{u(p)^2\left|q_p^T y\right|}}{\sqrt{\sum_{p' \in P} u(q)^2\left|q_{p'}^T y\right|}}$$

$$\le \sup_{y \in \mathbb{R}^{d+2}} \frac{\sqrt{u(p)^2\left|q_p^T y\right|}}{\sqrt{\sum_{p' \in P} u(q)^2\left|q_{p'}^T y\right|}}, \tag{12}$$

where the last inequality follows from the properties associated with the supremum operation.

Let $u' : P' \to [0,\infty)$ be a weight function such that for every $p \in P$, $u'(q_p) = u(p)^2$, $f : P' \times Y \to [0,\infty)$ be a function such that for every $q \in P'$ and $y \in Y$, $f(q,y) = \left|q^T y\right|$. Plugging in $P := P'$, $w := u$, $d := d+2$, and $f := f$ into Lemma 3 yields for every $p \in P$

$$\sup_{x \in X} \frac{w(p)f(p,x)}{\sum_{q \in P} w(q)f(q,x)} \le e^3\sqrt{\|q_p\|_2}\left(1+3\sqrt{\|q_p\|_2}\right)\left(\frac{u(p)}{\sum_{p' \in P} u(p')} + u(p)\sqrt{\|U(q_p)\|_1}\right). \tag{13}$$

Note that by definition, $q^* \in \arg\sup_{q \in P'} e^3\sqrt{\|q_p\|_2}\left(1+3\sqrt{\|q_p\|_2}\right)$. Then the total sensitivity is bounded by

$$\sum_{q \in P} \sup_{x \in X} \frac{w(p)f(p,x)}{\sum_{q \in P} w(q)f(q,x)} \le e^3\sqrt{\|q^*\|_2}\left(1+3\sqrt{\|q^*\|_2}\right)\left(1+\sqrt{n}(d+2)^{1.25}\right), \tag{14}$$

where the $\sqrt{n}$ follows from $\sum_{p' \in P} \sqrt{\|U(q_{p'})\|_1} \le \sqrt{n}\sqrt{\sum_{p' \in P} \|U(q_{p'})\|_1}$, which is used when using Lemma 3. This inequality is a result of Cauchy-Schwartz's inequality.

**Handling queries from $\mathbb{R}^d \setminus X$.** First, we observe that for any integer $m \ge 1$ and $x, y \in \mathbb{R}^m$,

$$-\|x\|_2 - \|y\|_2 \le -\|x-y\|_2 \le \|x\|_2 - \|y\|_2, \tag{15}$$

where the first inequality holds by the triangle inequality, and the second inequality follows from the reverse triangle inequality.

Thus, by letting $x_p \in \arg\sup_{x \in \mathbb{R}^d \setminus X} \frac{w(p)f(p,x)}{\sum_{q \in P} w(q)f(q,x)}$ for every $p \in P$, we obtain that

$$\frac{w(p)f(p,x_p)}{\sum_{q \in P} w(q)f(q,x_p)} \leq \frac{w(p)e^{\|p\|_2 - \|x_p\|_2}}{\sum_{q \in P} w(q)e^{-\|q\|_2 - \|x_p\|_2}} \leq \frac{w(p)e^{\|p\|_2 - \|x_p\|_2}}{\sum_{q \in P} w(q)e^{-\sqrt{\|q^*\|_2} - \|x_p\|_2}} = \frac{e^{\|p\|_2 + \sqrt{\|q^*\|_2}}w(p)}{\sum_{q \in P} w(q)},$$

(16)

where the first inequality holds by equation 15, and the second inequality holds since $\sqrt{\|q^*\|_2} \geq \|p\|_2$ for every $p \in P$.

Combining equation 9, equation 13, equation 14, and equation 16, yields that for every $p \in P$

$$s(p) \leq e^{3\sqrt{\|q_p\|_2}}\left(1 + 3\sqrt{\|q_p\|_2}\right)\left(\frac{u(p)}{\sum_{p' \in P} u(p')} + u(p)\sqrt{\|U(q_p)\|_1}\right) + \frac{e^{\|p\|_2 + \sqrt{\|q^*\|_2}}w(p)}{\sum_{q \in P} w(q)}$$

and

$$\sum_{p \in P} s(p) \leq 2e^{3\sqrt{\|q^*\|_2}} + e^{3\sqrt{\|q^*\|_2}}\left(1 + 3\sqrt{\|q^*\|_2}\right)\left(1 + \sqrt{n}(d+2)^{1.25}\right).$$

$\square$

**Theorem 7** (Coreset for the Laplacian loss function). *Let $(P, w, \mathbb{R}^d, f)$ be query space as in Definition 9 such that every $p \in P$ satisfies $\|p\|_2 \leq 1$. For $x \in \mathbb{R}^d$ and $p \in P$, let $f(p, x) := e^{-\|p-x\|_2}$. Let $\varepsilon, \delta \in (0, 1)$. Then there exists an algorithm which given $P, w, \varepsilon, \delta$ return a weighted set $(S, v)$ where $S \subseteq P$ of size $O\left(\frac{\sqrt{n}d^{1.25}}{\varepsilon^2}\left(\log n + \log d + \log\frac{1}{\delta}\right)\right)$ and a weight function $v : S \to [0, \infty)$ such that $(S, v)$ is an $\varepsilon$-coreset of $(P, w)$ with probability at least $1 - \delta$.*

*Proof.* Plugging in the query space $(P, w, \mathbb{R}^d, f)$ into Lemma 14, we obtain that a bound on the sensitivities $s(p)$ for every $p \in P$ and a bound on the total sensitivities $t := 2e^5 + 3e^5(1+5)\left(1 + \sqrt{n}(d+2)^{1.5}\right)$, since the $\max_{q \in P}\|q\|_2 \leq 1$. By plugging the bounds on the sensitivities, the bound on the total sensitivity $t$, probability of failure $\delta \in (0, 1)$, and approximation error $\varepsilon \in (0, 1)$ into Theorem 11, we obtain a subset $S' \subseteq Q$ and $v' : S' \to [0, \infty)$ such that the tuple $(S', v')$ is an $\varepsilon$-coreset for $(P, w)$ with probability at least $1 - \delta$. $\square$

### B.4 PROOF OF THEOREM 8

To prove Theorem 8, we first prove the following theorem.

**Theorem 15.** *There exists an algorithm that samples $O\left(\frac{e^{8R^2}R^2d^{1.5}}{\varepsilon^2}\left(R^2 + \log d + \log\frac{2}{\delta}\right)\right)$ points to form weighted sets $(S_1, w_1)$ and $(S_2, w_2)$ such that with probability at least $1 - 2\delta$,*

$$\frac{\left|\sum_{p \in P} y(p)\rho(\|p - c^{(i)}\|_2) - \left(\sum_{p \in S_1} w_1(p)e^{-\|p-c^{(i)}\|_2^2} + \sum_{q \in S_2} w_2(q)e^{-\|q-c^{(j)}\|_2^2}\right)\right|}{\left|\phi^+\left(c^{(i)}\right) + \phi^-\left(c^{(i)}\right)\right|} \leq \varepsilon.$$

*Proof.* We first construct strong coresets for both $\phi^+$ and $\phi^-$. If $\|c_i\|_2 \leq R$ for all $i \in [n]$, then by Theorem 6, it suffices to sample a weighted weight $S$ of size $O\left(\frac{e^{12R^2}R^2d^{1.5}}{\varepsilon^2}\left(R^2 + \log d + \log\frac{2}{\delta}\right)\right)$ to achieve a strong coreset $S_1$ with corresponding weighting function $w_1$ for $\phi^+$ with probability at least $1 - \frac{\delta}{2}$. Similarly, we obtain a strong coreset for $\phi^-$ with probability at least $1 - \frac{\delta}{2}$ by sampling a set $S_2$ with weights $w_2$ of size $O\left(\frac{e^{8R^2}R^2d^{1.5}}{\varepsilon^2}\left(R^2 + \log d + \log\frac{2}{\delta}\right)\right)$.

Hence by the definition of a strong coreset, we have that

$$\left| \sum_{\substack{p \in P \\ y(p) > 0}} y(p)\rho(\|p - c^{(i)}\|_2) - \sum_{p \in S_1} w_1(p)\rho(\|p - c^{(i)}\|_2) \right| \le \varepsilon \sum_{\substack{p \in P \\ y(p) > 0}} y(p)\rho(\|p - c^{(i)}\|_2)$$

$$= \varepsilon \phi^+(c^{(i)})$$

and

$$\left| \sum_{\substack{p \in P \\ y(p) < 0}} y(p)\rho(\|p - c^{(i)}\|_2) - \sum_{q \in S_2} w_2(q)\rho(\|q - c^{(i)}\|_2) \right| \le \varepsilon \sum_{\substack{p \in P \\ y(p) < 0}} |y(p)|\rho(\|p - c^{(i)}\|_2)$$

$$= \varepsilon \phi^-(c^{(i)})$$

for any input $\mathbf{x} \in \mathbb{R}^n$.

Thus by triangle inequality and a slight rearrangement of the inequality, we have that with probability at least $1 - \delta$,

$$\frac{\left| \sum_{p \in P} y(p)\rho(\|p - c^{(i)}\|_2) - \left( \sum_{p \in S_1} w_1(p)e^{-\|p - c^{(i)}\|_2^2} + \sum_{q \in S_2} w_2(q)e^{-\|q - c^{(j)}\|_2^2} \right) \right|}{\left| \phi^+\left(c^{(i)}\right) + \phi^-\left(c^{(i)}\right) \right|} \le \varepsilon.$$

$\square$

To prove Theorem 8, we split $\alpha_i$ into the sets $\{i | \alpha_i > 0\}$ and $\{i | \alpha_i < 0\}$ and apply Theorem 15 to each of the sets. We emphasize that this argument is purely for the purposes of analysis so that the algorithm itself does not need to partition the $\alpha_i$ quantities (and so the algorithm does not need to recompute the coreset when the values of the weights $\alpha_i$ change over time). We thus get the following guarantee:

**Theorem 8.** *There exists an algorithm that samples* $O\left( \frac{e^{8R^2} R^2 d^{1.5}}{\varepsilon^2} \left( R^2 + \log d + \log \frac{2}{\delta} \right) \right)$ *points to form weighted sets* $(S_1, w_1)$ *and* $(S_2, w_2)$ *such that with probability at least* $1 - 2\delta$,

$$\frac{\left| \sum_{p \in P} y(p)\phi(p) - \left( \sum_{\substack{i \in [L] \\ \alpha_i > 0}} \alpha_i \sum_{p \in S_1} w_1(p)e^{-\|p - c^{(i)}\|_2^2} + \sum_{\substack{j \in [L] \\ \alpha_j < 0}} \alpha_j \sum_{q \in S_2} w_2(q)e^{-\|q - c^{(j)}\|_2^2} \right) \right|}{\left| \sum_{\substack{i \in [L] \\ \alpha_i > 0}} \alpha_i \left( \phi^+\left(c^{(i)}\right) + \phi^-\left(c^{(i)}\right) \right) \right| + \left| \sum_{\substack{i \in [L] \\ \alpha_i < 0}} |\alpha_i| \left( \phi^+\left(c^{(i)}\right) + \phi^-\left(c^{(i)}\right) \right) \right|} \le \varepsilon.$$

## C  LOWER BOUND ON THE CORESET SIZE FOR THE GAUSSIAN LOSS FUNCTION - ILLUSTRATION

Here, we illustrate one dataset such that for any approximation $\varepsilon$, the $\varepsilon$-coreset must contain at least half the points to ensure the desired approximation from a theoretical point of view.

## D  EXPERIMENTAL RESULTS - EXTENDED

### D.1  MNIST RESULTS

In what follows, we present our subset selection results on the MNIST dataset at Table 4.

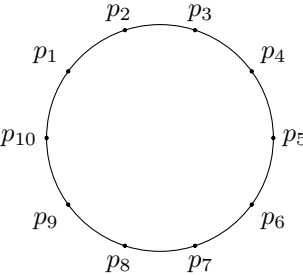

Figure 3: Evenly distributed points on some circle where the minimal distance between each point and any other point is at least $\sqrt{\ln n}$, where in this example $n = 10$.

Table 4: Data Selection Results for MNIST using LeNet

| | Top-1 Test accuracy of the Model(%) | | | | Model Training time(in hrs) | | | |
|---|---|---|---|---|---|---|---|---|
| Budget(%) | 1% | 3% | 5% | 10% | 1% | 3% | 5% | 10% |
| FULL (skyline for test accuracy) | 99.35 | 99.35 | 99.35 | 99.35 | 0.82 | 0.82 | 0.82 | 0.82 |
| RANDOM (skyline for training time) | 94.55 | 97.14 | 97.7 | 98.38 | 0.0084 | 0.03 | 0.04 | 0.084 |
| RANDOM-WARM (skyline for training time) | 98.8 | 99.1 | 99.1 | 99.13 | 0.0085 | 0.03 | 0.04 | 0.085 |
| GLISTER | 93.11 | 98.062 | 99.02 | 99.134 | 0.045 | 0.0625 | 0.082 | 0.132 |
| GLISTER-WARM | 97.63 | 98.9 | 99.1 | 99.15 | 0.04 | 0.058 | 0.078 | 0.127 |
| CRAIG | 96.18 | 96.93 | 97.81 | 98.7 | 0.3758 | 0.4173 | 0.434 | 0.497 |
| CRAIG-WARM | 98.48 | 98.96 | 99.12 | 99.14 | 0.2239 | 0.258 | 0.2582 | 0.3416 |
| CRAIGPB | 97.72 | 98.47 | 98.79 | 99.05 | 0.08352 | 0.106 | 0.1175 | 0.185 |
| CRAIGPB-WARM | 98.47 | 99.08 | 99.01 | 99.16 | 0.055 | 0.077 | 0.0902 | 0.1523 |
| GRADMATCH | 98.954 | 99.174 | 99.214 | 99.24 | 0.05 | 0.0607 | 0.097 | 0.138 |
| GRADMATCH-WARM | 98.86 | 99.22 | 99.28 | 99.29 | 0.046 | 0.057 | 0.089 | 0.132 |
| GRADMATCHPB | 98.7 | 99.1 | 99.25 | 99.27 | 0.04 | 0.051 | 0.07 | 0.11 |
| GRADMATCHPB-WARM | **99.0** | **99.23** | 99.3 | 99.31 | 0.038 | 0.05 | 0.065 | 0.10 |
| RBFNN CORESET (OURS) | 98.98 | 99.2 | **99.31** | **99.32** | 0.028 | 0.051 | 0.062 | 0.098 |

## D.2 STANDARD DEVIATION AND STATISTICAL SIGNIFICANCE RESULTS

Tables 5–7 show the standard deviation results over five training runs on CIFAR10, CIFAR100, and MNIST datasets, respectively.

Table 5: Data Selection Results for CIFAR10 using ResNet-18: Standard deviation of the Model (for 5 runs)

| | Standard deviation of the Model(for 5 runs) | | | |
|---|---|---|---|---|
| Budget(%) | 5% | 10% | 20% | 30% |
| FULL (skyline for test accuracy) | 0.032 | 0.032 | 0.032 | 0.032 |
| RANDOM (skyline for training time) | 0.483 | 0.518 | 0.524 | 0.538 |
| RANDOM-WARM (skyline for training time) | 0.461 | 0.348 | 0.24 | 0.1538 |
| GLISTER | 0.453 | 0.107 | 0.046 | 0.345 |
| GLISTER-WARM | 0.325 | 0.086 | 0.135 | 0.129 |
| CRAIG | 0.289 | 0.2657 | 0.1894 | 0.1647 |
| CRAIG-WARM | 0.123 | 0.1185 | 0.1058 | 0.1051 |
| CRAIGPB | 0.152 | 0.1021 | 0.086 | 0.064 |
| CRAIGPB-WARM | 0.0681 | 0.061 | 0.0623 | 0.0676 |
| GRADMATCH | 0.192 | 0.123 | 0.112 | 0.1023 |
| GRADMATCH-WARM | 0.1013 | 0.1032 | 0.091 | 0.1034 |
| GRADMATCHPB | 0.0581 | 0.0571 | 0.0542 | 0.0584 |
| GRADMATCHPB-WARM | 0.0542 | 0.0512 | 0.0671 | 0.0581 |
| RBFNN CORESET (OURS) | 0.25 | 0.2 | 0.17 | 0.13 |
| RBFNN CORESET-WARM (OURS) | 0.21 | 0.16 | 0.13 | 0.12 |

Table 6: Data Selection Results for CIFAR100 using ResNet-18: Standard deviation of the Model (for 5 runs)

| Budget(%) | Standard deviation of the Model(for 5 runs) | | | |
| --- | --- | --- | --- | --- |
| | 5% | 10% | 20% | 30% |
| FULL (skyline for test accuracy) | 0.051 | 0.051 | 0.051 | 0.051 |
| RANDOM (skyline for training time) | 0.659 | 0.584 | 0.671 | 0.635 |
| RANDOM-WARM (skyline for training time) | 0.359 | 0.242 | 0.187 | 0.175 |
| GLISTER | 0.463 | 0.15 | 0.061 | 0.541 |
| GLISTER-WARM | 0.375 | 0.083 | 0.121 | 0.294 |
| CRAIG | 0.3214 | 0.214 | 0.195 | 0.187 |
| CRAIG-WARM | 0.18 | 0.132 | 0.125 | 0.115 |
| CRAIGPB | 0.12 | 0.134 | 0.123 | 0.115 |
| CRAIGPB-WARM | 0.1176 | 0.1152 | 0.1128 | 0.111 |
| GRADMATCH | 0.285 | 0.176 | 0.165 | 0.156 |
| GRADMATCH-WARM | 0.140 | 0.134 | 0.142 | 0.156 |
| GRADMATCHPB | 0.104 | 0.111 | 0.105 | 0.097 |
| GRADMATCHPB-WARM | 0.093 | 0.101 | 0.100 | 0.098 |
| RBFNN CORESET (OURS) | 0.3 | 0.19 | 0.18 | 0.16 |
| RBFNN CORESET-WARM (OURS) | 0.19 | 0.14 | 0.11 | 0.1 |

Table 7: Data Selection Results for MNIST using LeNet: Standard deviation of the Model (for 5 runs)

| Budget(%) | Standard deviation of the Model(for 5 runs | | | |
| --- | --- | --- | --- | --- |
| | 1% | 3% | 5% | 10% |
| FULL (skyline for test accuracy) | 0.012 | 0.012 | 0.012 | 0.012 |
| RANDOM (skyline for training time) | 0.215 | 0.265 | 0.224 | 0.213 |
| RANDOM-WARM (skyline for training time) | 0.15 | 0.121 | 0.110 | 0.103 |
| GLISTER | 0.256 | 0.218 | 0.145 | 0.128 |
| GLISTER-WARM | 0.128 | 0.134 | 0.119 | 0.124 |
| CRAIG | 0.186 | 0.178 | 0.162 | 0.125 |
| CRAIG-WARM | 0.0213 | 0.0223 | 0.0196 | 0.0198 |
| CRAIGPB | 0.021 | 0.0209 | 0.0216 | 0.0204 |
| CRAIGPB-WARM | 0.023 | 0.0192 | 0.0212 | 0.0184 |
| GRADMATCH | 0.156 | 0.128 | 0.135 | 0.12 |
| GRADMATCH-WARM | 0.087 | 0.084 | 0.0896 | 0.0815 |
| GRADMATCHPB | 0.0181 | 0.0163 | 0.0147 | 0.0129 |
| GRADMATCHPB-WARM | 0.0098 | 0.012 | 0.0096 | 0.0092 |
| RBFNN CORESET (OURS) | 0.18 | 0.13 | 0.11 | 0.1 |
| RBFNN CORESET-WARM (OURS) | 0.09 | 0.08 | 0.05 | 0.01 |

## D.3  RBF FITTING - OUR CORESET VERSUS UNIFORM SAMPLING

In this experiment, we aim to show the efficacy of our coreset against uniform sampling for the task of fitting RBFs. Specifically speaking, we have taken two datasets: (i) 3D spatial networks (Dua et al., 2017) , and (ii) HTRU2 pulsar dataset (Dua et al., 2017) .

In the following figures, we report the averaged approximation factor over 20 trials, where the shaded area corresponds to the median absolute deviation, a more robust statistical measurement that the standard deviation.

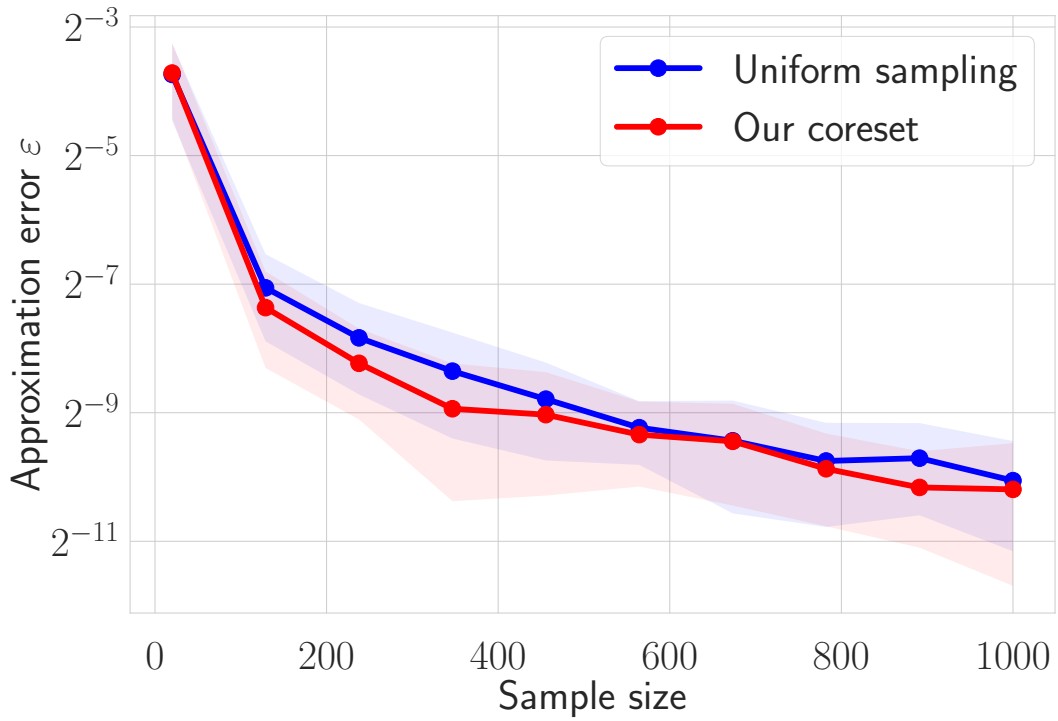

Figure 4: The averaged approximation error: The $x$-axis is the size of the chosen subset, while the $y$ axis is the averaged approximation error. This is with respect to dataset (i).

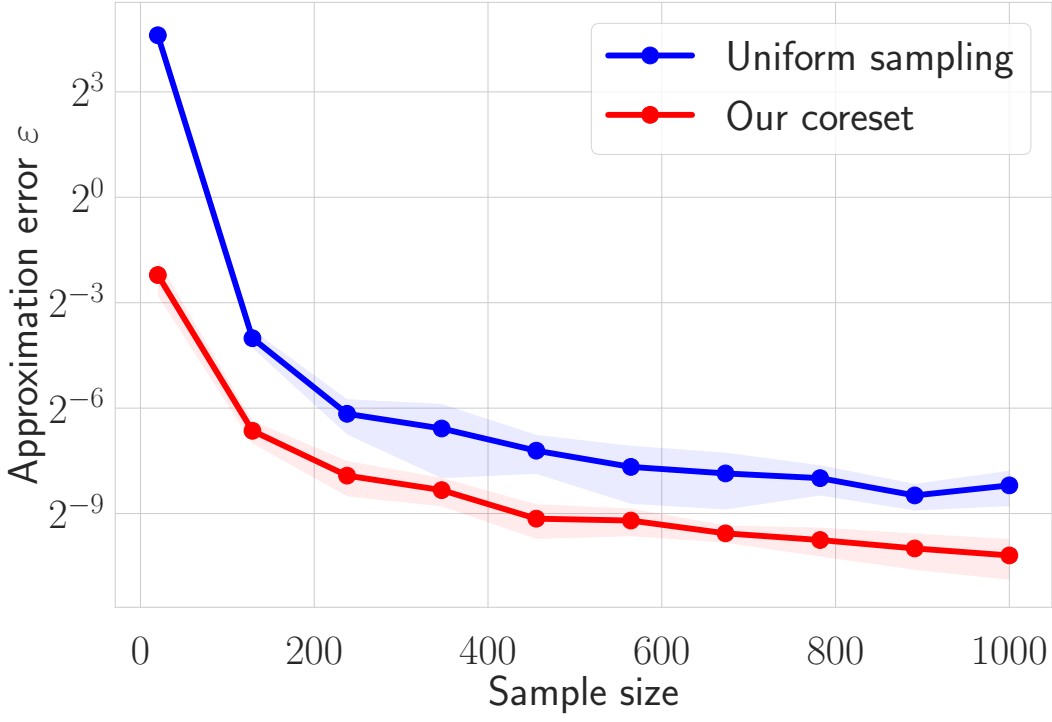

Figure 5: The averaged approximation error: The $x$-axis is the size of the chosen subset, while the $y$ axis is the averaged approximation error. This is with respect to dataset (ii).

