# OpenReview forum: "Approximating any Function via Coreset for Radial Basis Functions: Towards Provable Data Subset Selection For Efficient Neural Networks training"
_ICLR.cc/2023/Conference — Submitted to ICLR 2023_

### Official Review · Reviewer_zVhj · 2022-10-24

**Confidence:** 2
**Correctness:** 2
**Technical Novelty And Significance:** 2
**Empirical Novelty And Significance:** 3
**Recommendation:** 5

**Clarity, Quality, Novelty And Reproducibility:**

Quality: This paper is not very technically sound.

Clarity: This paper is well organized. I find it easy to follow.

Significance: I think the results in this paper are not very significant.

**Strength And Weaknesses:**

Although the experiment results seem promising, the technical parts may not seem very solid, which makes me have some concerns about the correctness of the results. For example,

1. In Theorems 4 and 6 (and somewhere else), the authors write something like $f(n)\in O(n)$, this is not the standard notation, it should always be $f(n)= O(n)$ in mathematics.

2. In Theorem 4, actually the author prove that $\sum s(p) \geq n/2$. In this case, the results should be $\sum s(p) = \Omega (n)$ (the big Omega notation), not $\sum s(p) \in O (n)$.

3. The formula in Claim 5 is not correct since letting $R=1$ will give $e/2\leq 1$, this can not be true.

4. The right hand side of formula in Claim 5 could be $1$, since $e^{-|p^T \cdot x|} \leq 1$. I don't understand why it writes $e^{-|p^T \cdot x|} \leq 1 + |p^T \cdot x|$.

**Summary Of The Paper:**

The authors introduced a coreset that provably approximates any function that can be represented by an RBFNN architecture, and use the coreset to suggest a provable data subset selection algorithm for training deep neural networks. The better performances of their algorithms are verified by some experiments.



**Summary Of The Review:**

Due to many confusions in the technical parts, as explained above, I tend not to accept this paper.

---

> ### Author Response · Authors · 2022-11-13
> **Response to Reviewer zvhj**
>
> **We thank the reviewer for the time and effort in reviewing our paper and greatly appreciate the questions raised, the constructive criticism of our work, and the knowledgeable comments. Your insightful comments are very much appreciated.**
>
> **Comment 1:** In Theorems 4 and 6 (and somewhere else), the authors write something like $f(n)\in O(n)$, this is not the standard notation, it should always be $f(n)=O(n)$ in mathematics.
>
> **Answer:** Yes, we agree with this notation and have facilitated it throughout the paper in the updated version.
>
> ---------------------------------------------------------------------------------------------------------------------------------------------------------------------------------------
>
> **Comment 2:** In Theorem 4, actually the author prove that $\sum s(p)\ge n/2$ . In this case, the results should be $\sum s(p)=\Omega(n)$ (the big Omega notation), not $\sum s(p)\in O(n)$.
>
> **Answer:** Indeed, the main point of Theorem 4 is to show the lower bound that without any distributional assumptions, there exists a set of points of size $n$ such that any coreset construction based on sensitivity sampling requires at least $\frac{n}{2}=\Omega(n)$ points. We have fixed this typo in the updated version, thanks for pointing it out.
>
> ---------------------------------------------------------------------------------------------------------------------------------------------------------------------------------------
>
> **Comment 3:** The formula in Claim 5 is not correct since letting $R=1$ will give $e/2\le 1$, this can not be true.
>
> **Answer:** Yes, we remark that the proof of Claim 5 in the appendix proves that $\frac{1}{e^R(1+R)}(1+|p^Tx|)\leq e^{-|p^Tx|}$. That is, the term in the denominator is $e^R$ rather than $e^{-R}$. We have corrected this term in the statement of Claim 5, thanks for pointing it out!
>
> ---------------------------------------------------------------------------------------------------------------------------------------------------------------------------------------
>
> **Comment 4:** The right hand side of formula in Claim 5 could be $1$, since $e^{-|p^T\cdot x|}\leq 1$. I don't understand why it writes $e^{-|p^T \cdot x|} \leq 1+|p^T \cdot x|$.
>
> **Answer:** The idea behind this bound is to move the problem from the RBF loss function to a variant of the $\ell_1$-regression loss function. By doing so, we are able to upper-bound the sensitivity of the points using the upper bounds for the sensitivity of the points in the context of the $\ell_1$-regression problem.
>
> ---------------------------------------------------------------------------------------------------------------------------------------------------------------------------------------
>
> **Comment 5:** Quality: This paper is not very technically sound.
>
> **Answer:** We thank the reviewer for the professional criticism. In light of your comments, the typo in the statement of Claim 5 has been fixed, while the proofs remained the same since the typo was not incorporated in them. We hope now that the reviewer will see that the correctness is complete and that no more typos are involved.
>
> ---------------------------------------------------------------------------------------------------------------------------------------------------------------------------------------
>
> **Comment 6:** Significance: I think the results in this paper are not very significant.
>
> **Answer:** We appreciate the reviewer's concern. We have addressed all the aforementioned typos in our paper, thus now the paper is theoretically complete. Note that, our paper is the first to address coresets for RBFNNs, and our coreset is generic in the sense that it can approximate multiple cost functions contrary to previous results. We hope this paper will be referred to as a stepping stone in the research community toward building stronger and better coresets in this line of work
>
> As for the experimental section, we are currently conducting experiments handling the ImageNet dataset. We are also conducting experiments that aim to show the efficacy of our coreset in fitting RBFs. Finally, we have added better results for cifar10 and cifar100 when applying the warm-variant to our methods; see updated tables.  If you have any additional experiments in mind, we will be happy to incorporate them into our paper.

---

> > ### Comment · Reviewer_zVhj · 2022-11-17
> > **Thanks for the reply**
> >
> > Thanks for the detailed reply from the authors, which solves many of my questions. I will keep the current score and be neutral about the acceptance of this paper.

---

### Official Review · Reviewer_NHbT · 2022-10-25

**Confidence:** 2
**Correctness:** 3
**Technical Novelty And Significance:** 2
**Empirical Novelty And Significance:** 2
**Recommendation:** 6

**Clarity, Quality, Novelty And Reproducibility:**

I believe that the idea of selecting the coreset with RBFNN is interesting and novel. Moreover, the paper is generally well-written although some parts of the paper are not easy-to-follow due to organization. I believe that the quality of the paper meets the standard of the ICLR.

The authors did not provide a code or details of the experiment set-up and I believe that it would be difficult to reproduce the experiments at the current state.

Below, I left some additional comments:
- I found the description in Section 1 - coresets to be confusing at first. A more description of the query selection before showing the equation would be useful.
- In addition, I think that the reorganization of the materials presented in Section 1 would be helpful. At the moment, the motivation for the work is not clear when first reading the introduction.
- While I believe that Figure 1 is extremely useful to capture the core contribution, there needs to be a detailed description. At the moment, the main text does not mention Figure 1.
- In my current understanding, the algorithm proposed in the paper could be split into two parts: RBFNNs and coreset selection. Would it be possible to set up an experiment that confirms the accuracy of each method (in step)?

Minor comments:
- While I believe that Figure 1 is extremely useful to capture the core contribution, there needs to be
In Section 5, “... after each gradient update to support there theoretic…” →“... after each gradient update to support their theoretic…”.
- There are several places where \citet and \citep are misused (e.g., Section 6, competing methods).
- In Table 1, it would be helpful if the fastest model training time is bolded. The same applies to Table 2 and Table 3.
- The resolution for figure 3 is bad.


**Strength And Weaknesses:**

As detailed below, the paper is well-written and I believe that the idea presented in this paper is novel. Note that I am not familiar with the background and did not fully understand the mathematical justification. Hence, my most concerns lie in the experiment section:

- While the paper is generally easy-to-follow, the motivations in Section 1 are unclear.
- The paper does not sufficiently describe the limitation of the proposed approach. What are additional hyperparameters that need to be tuned (e.g., $R$) and how robust are they? The paper briefly mentions that the algorithm is robust to these hyperparameters, but did not empirically verify this.
- The performance seems to be limited for CIFAR-100 and MNIST datasets.
- The paper does not describe the experiment details and it would be challenging to reproduce the experiment. Moreover, the details on how the hyperparameters (e.g., learning rate) are selected are missing.
- The evaluations were done on three relatively small datasets and architectures (e.g., MNIST, CIFAR-10, and CIFAR-100) and it is unclear if the method will scale to large settings.

**Summary Of The Paper:**

The paper proposes an algorithm for coreset selection with low computation time. The key idea is to approximate a function using RBFNN on a large dataset and then construct coresets for radial basis and Laplicain loss function. Empirically, the authors show that the proposed method can find a small coreset with competitive performance with the existing baseline methods on MNIST, CIFAR-10, and CIFAR-100 datasets.


**Summary Of The Review:**

Overall, I believe that the idea presented in the paper is interesting and the paper is overall well-written. There are some concerns about the experiment analysis (e.g., the small number of baseline methods, evaluation on relatively small datasets and architectures, limited performance on CIFAR-100 and MNIST, and no ablation studies). I recommend 6.

---

> ### Author Response · Authors · 2022-11-13
> **Response to Reviewer NHbT: Part III**
>
> **Comment 11:** In Table 1, it would be helpful if the fastest model training time is bolded. The same applies to Table 2 and Table 3.
>
> **Answer:** We thank the reviewer for improving the readability of our results. The paper has been updated accordingly.
>
> ---------------------------------------------------------------------------------------------------------------------------------------------------------------------------------------
>
> **Comment 12:** The resolution for figure 3 is bad.
>
> **Answer:** We thank the reviewer for his comment. We have enlarged this figure to make it more clear. In addition, we are conducting an additional experiment that will lead to an additional figure highlighting our coreset’s efficacy in approximating the output of the RBF fitting problem which is one of the core contributions on which our coreset for RBFNNs relies.

---

> ### Author Response · Authors · 2022-11-13
> **Response to Reviewer NHbT: Part II**
>
> **Comment 5:** I found the description in Section 1 - coresets to be confusing at first. A more description of the query selection before showing the equation would be useful.
>
> **Answer:** We thank the reviewer for his fruitful suggestion. We have modified the introduction of our paper for more clarity.
>
> ---------------------------------------------------------------------------------------------------------------------------------------------------------------------------------------
>
> **Comment 6:** In addition, I think that the reorganization of the materials presented in Section 1 would be helpful. At the moment, the motivation for the work is not clear when first reading the introduction.
>
> **Answer:** This is indeed important, thanks for pointing this out. The motivation for these contributions is that without significant data subset selection, the training time may be prohibitive for deep neural networks, on a large training dataset.  To do so, we explore radial basis function neural networks (RBFNNs) that are known for their ability to estimate any continuous function on a closed bounded set with arbitrary precision. We propose the first coreset construction algorithm for RBFNNs, which is to compute a small weighted subset that roughly approximates the input data loss on any radial basis function neural network of this size and thereby roughly approximates any function defined (approximated) by such an RBFNN on the large input data.
>
> Then we apply our novel coreset constructions to prove an efficient model-independent subset selection algorithm for training neural networks. We have increased the discussion for the motivation in Section 1 of the updated version, thanks for the suggestion.
>
> ---------------------------------------------------------------------------------------------------------------------------------------------------------------------------------------
>
> **Comment 7:** While I believe that Figure 1 is extremely useful to capture the core contribution, there needs to be a detailed description. At the moment, the main text does not mention Figure 1.
>
> **Answer:** We thank the reviewer for pointing this out. We have incorporated this in the updated paper.
>
> ---------------------------------------------------------------------------------------------------------------------------------------------------------------------------------------
>
> **Comment 8:**  In my current understanding, the algorithm proposed in the paper could be split into two parts: RBFNNs and coreset selection. Would it be possible to set up an experiment that confirms the accuracy of each method (in step)?
>
> **Answer:** We thank the reviewer for the fruitful suggestion.  Figure 3 presents the results of function approximation where both the RBFNN training and the coreset construction are applied.  In light of this, we will add an additional figure to emphasize the quality of our coreset with respect to the RBF loss function which our contributions rely on.
>
> As for our experiments involving subset selection for boosting the training of deep neural models, we did not require the training of RBFNNs due to the fact that our coreset’s provable guarantees hold for any query in a ball of radius $R$ (theoretically speaking).
>
> ---------------------------------------------------------------------------------------------------------------------------------------------------------------------------------------
>
> **Comment 9:** While I believe that Figure 1 is extremely useful to capture the core contribution, there needs to be In Section 5, “... after each gradient update to support there theoretic…” →“... after each gradient update to support their theoretic…”.
>
> **Answer:** We thank the reviewer for raising this improvement. The paper has been updated.
>
> ---------------------------------------------------------------------------------------------------------------------------------------------------------------------------------------
>
> **Comment 10:** There are several places where \citet and \citep are misused (e.g., Section 6, competing methods).
>
> **Answer:** We thank the reviewer for the keen observation. The paper has been updated accordingly.

---

> ### Author Response · Authors · 2022-11-13
> **Response to Reviewer NHbT: Part I**
>
> **We would like to thank the reviewer for the professional reviewer, and the highly knowledgeable comments to help improve our manuscript. We indeed appreciate your time and efforts.**
>
> **Comment 1:** While the paper is generally easy-to-follow, the motivations in Section 1 are unclear.
>
> **Answer:** Thanks for pointing this out. Our main contributions are twofold. 1) We introduce the first coreset construction for RBFNNs, thus introducing a data subset selection that provably approximates any function that can be approximated by a given parametrized RBFNN, which is especially important because RBFNNs are universal approximators in the sense that with a sufficient number of hidden neurons, they can approximate any continuous function (on a closed, bounded subset of $R^d$) with arbitrary precision. 2) We apply our novel coreset constructions to prove an efficient model-independent subset selection algorithm for training neural networks.
>
> The motivation for these contributions is that without significant data subset selection, the training time may be prohibitive for RBFNNs with a large number of layers or a large training dataset. We have increased the discussion for motivation in Section 1 of the updated version.
>
> ---------------------------------------------------------------------------------------------------------------------------------------------------------------------------------------
>
> **Comment 2:** The paper does not sufficiently describe the limitation of the proposed approach. What are additional hyperparameters that need to be tuned (e.g., $R$) and how robust are they? The paper briefly mentions that the algorithm is robust to these hyperparameters, but did not empirically verify this.
>
> **Answer:** Theoretically speaking, the size of the coreset is parametrized by the value $R$, which upper bounds the largest magnitude of any point in the query space. In particular, Theorem 4 shows that without assumptions on $R$, there provably does not exist any sublinear size coreset construction for the Gaussian loss function (and thus RBFs).
>
> As for the experiments, $R$ here refers to the number of iterations until a new coreset is constructed. In this context, and in our experiments, we adopted the same exact settings of “GRAD-MATCH: gradient matching based data subset selection for efficient deep model training” by KrishnaTeja Killamsetty, Durga Sivasubramanian, Ganesh Ramakrishnan, Abir De, and Rishabh K. Iyer. This is done to ensure fairness across our experiments.
>
> ---------------------------------------------------------------------------------------------------------------------------------------------------------------------------------------
>
> **Comment 3:** The performance seems to be limited for CIFAR-100 and MNIST datasets. | The evaluations were done on three relatively small datasets and architectures (e.g., MNIST, CIFAR-10, and CIFAR-100) and it is unclear if the method will scale to large settings.
>
> **Answer:** We appreciate your comment. We are conducting experiments with respect to the ImageNet dataset. We will update the paper once the results are done, and we will update you here.
>
> ---------------------------------------------------------------------------------------------------------------------------------------------------------------------------------------
>
> **Comment 4:** The authors did not provide a code or details of the experiment set-up and I believe that it would be difficult to reproduce the experiments at the current state. | The paper does not describe the experiment details and it would be challenging to reproduce the experiment. Moreover, the details on how the hyperparameters (e.g., learning rate) are selected are missing.
>
> **Answer:** We thank the reviewer for raising these concerns. We refer the reviewer to the paragraphs named “The setting” and ”Subset sizes and the R parameter", in Section 6. Here most of the hyperparameters were detailed in the submitted version. Following this comment, we modified the name from “The setting” to “The setting and hyperparameters selection”, to make it more informative.
> Finally, we will upload the code responsible for our subset selection/coreset generation.

---

### Official Review · Reviewer_A6tT · 2022-10-25

**Confidence:** 2
**Clarity, Quality, Novelty And Reproducibility:** Good
**Correctness:** 3
**Technical Novelty And Significance:** 2
**Empirical Novelty And Significance:** 2
**Recommendation:** 6

**Strength And Weaknesses:**

Strength

- This paper develops the first coreset construction algorithm for RBFNNs and Laplacian cost function. These properties of RBFNNs are further leveraged to approximate the gradients of any deep neural networks.

- The authors also develop provable theoretical guarantees for the proposed coreset construction algorithm.

Questions:

- In the abstract, `notoriously known' should be `well known'

- The authors study RBFNNs and use them to help data selection as they can approximate any continuous function on a closed bounded set with arbitrary precision given enough hidden neurons. But many other neural networks are also universal approximators, I am wondering what is the advantage to use RBFNNs and bridge them with general deep neural networks.

- In the experiments, the authors compare with some SOTA baselines with warm start, i.e., training on the whole data for 50% of the training time before training the other 50% on the coreset. It is shown that the proposed algorithm outperforms these SOTAs without warm start, but worse than with warm start. As warm start is a good strategy used in practice, I suggest the author also add the proposed algorithm + warm start to the comparison.

- How does this approach compare to the clustering-based data selection as discussed in the recent work https://arxiv.org/pdf/2206.14486.pdf

**Summary Of The Paper:**

In this paper, the authors propose the first coreset construction algorithm for RBFNNs, i.e., a small weighted subset which approximates the loss of the input data on any radial basis function network. This is achieved by constructing coresets for radial basis and Laplacian loss functions. The coreset is then used to develop a provable data subset selection algorithm for training deep neural networks, since the coreset approximates every function. Experimental results on function approximation and dataset subset selection on popular network architectures and data sets are presented to demonstrate the effectiveness of the algorithm in certain cases.

**Summary Of The Review:**

see above

---

> ### Author Response · Authors · 2022-11-13
> **Response to Reviewer A6tT**
>
> **We thank the reviewer for the positive feedback, professional review, and constructive comments. We greatly appreciate the time and effort of the reviewer to help us improve our work.**
>
> **Comment 1:** In the abstract, notoriously known' should be well known'
>
> **Answer:** We thank the reviewer for pointing this out. The paper has been updated accordingly.
>
> -------------------------------------------------------------------------------------------------------------------------------------------------------------------------------------------
>
> **Comment 2:** The authors study RBFNNs and use them to help data selection as they can approximate any continuous function on a closed bounded set with arbitrary precision given enough hidden neurons. But many other neural networks are also universal approximators, I am wondering what is the advantage to use RBFNNs and bridge them with general deep neural networks.
>
> **Answer:** The main advantage which we utilized in this paper is that RBFNN relies on fitting a linear combination of RBF loss functions. Hence, the simplicity of such neural networks (having one hidden layer with well-defined losses) motivated us to derive coreset construction for the RBFs and draw the theoretical guarantees that are needed to handle a linear combination of such functions.
>
> We note that, to the best of our knowledge, this is the first paper that addresses the problem of coreset construction for approximating any function that can be approximated by RBFNNs. Note that our coreset in a sense is universal. In other words, our coreset can be computed once and used to approximate a variety of functions using RBFNNs.
>
> We hope that our paper will be related as the first stepping stone in the field of boosting the training of neural networks that are universal approximators via coresets/subset selection.
>
> ---------------------------------------------------------------------------------------------------------------------------------------------------------------------------------------
>
> **Comment 3:** In the experiments, the authors compare with some SOTA baselines with warm start, i.e., training on the whole data for 50% of the training time before training the other 50% on the coreset. It is shown that the proposed algorithm outperforms these SOTAs without warm start, but worse than with warm start. As warm start is a good strategy used in practice, I suggest the author also add the proposed algorithm + warm start to the comparison.
>
> **Answer:** Following your comment, we have conducted more experiments with a warm start before applying our method, resulting in better results (see the updated tables in the paper). Thank you so much for this suggestion as it helped us improve our results!
>
> ---------------------------------------------------------------------------------------------------------------------------------------------------------------------------------------
>
> **Comment 4:** How does this approach compare to the clustering-based data selection as discussed in the recent work https://arxiv.org/pdf/2206.14486.pdf
>
> **Answer:** Our work focuses on data subset selection algorithms for RBFNNs through coreset constructions via sensitivity sampling. By comparison, the reference [SGSGM22] at https://arxiv.org/pdf/2206.14486.pdf performs a systematic study on data pruning algorithms for perceptrons. Thus although our goals are similar (and indeed the problem of training dataset reduction is well-motivated), the subject of our studies differ greatly. In particular, we remark that RBFNNs enjoy advantages such as simplicity of analysis, faster training time, and interpretability, compared to alternatives such as CNNs and perceptrons. We will add this discussion to our related work (along with other references that perform data pruning after the training stage).
>
> [SGSGM22] Ben Sorscher, Robert Geirhos, Shashank Shekhar, Surya Ganguli, Ari S. Morcos: Beyond neural scaling laws: beating power law scaling via data pruning. CoRR abs/2206.14486 (2022)

---

### Official Review · Reviewer_zk6H · 2022-10-25

**Confidence:** 3
**Clarity, Quality, Novelty And Reproducibility:** The paper is well-written and the pro…
**Correctness:** 4
**Technical Novelty And Significance:** 3
**Empirical Novelty And Significance:** 3
**Recommendation:** 6

**Strength And Weaknesses:**

Strength:

1. The authors propose a coreset construction algorithm for the RBF and Laplacian cost functions, which is efficient and in theory can be used to approximate the gradients of any deep neural networks.

2. The proposed algorithm construction leverages existing theoretical results; and the authors provide further theoretical analysis for RBF and Laplacian loss functions.

3. The proposed algorithm is efficient - the coreset can be computed before each epoch with negligible time by sampling according to the sensitivity distribution.

4. Extensive empirical evaluation that demonstrates the efficiency and effectiveness of the proposed algorithm.

Questions and weaknesses:

1. I'm curious about the experiment results - from Tables 1 and 2, the proposed RBFNN Coreset seems to be inferior to some other methods (e.g., GradMatchPB-WARM) for both accuracy and training time. Is this difference mainly coming from the warm start? If so, would warm start help to improve the performance of the RBFNN Coreset?

2. Why is the training time with a warm start shorter than the one without (e.g., Table 1)? Shouldn't train on the whole dataset increase the training time?

3. The standard deviation of the empirical results is not reported - I'm not fully convinced the differences are statistically significant.

Minor issue:

In definition 2 where the function $U(p)$ is defined, the $P$ should be $p$?

**Summary Of The Paper:**

In this paper, the authors propose a coreset construction algorithm for RBFNNs. The coreset corresponds to a data subset selection algorithm for training deep neural networks. Such a coreset can provably approximate any function that can be approximated by a given RBFNN. The proposed algorithm is efficient and the results are empirically verified.

**Summary Of The Review:**

This paper presents an interesting coreset construction for RBFNNs, and the authors provide solid theoretical analysis and empirical evaluation for the proposed algorithm.

---

> ### Author Response · Authors · 2022-11-13
> **Response to Reviewer zk6H**
>
> **We thank the reviewer for the professional review, careful reading, and clear detailed comments. Your insightful review has undoubtedly aided us in improving our paper.**
>
> **Comment 1:** I'm curious about the experiment results - from Tables 1 and 2, the proposed RBFNN Coreset seems to be inferior to some other methods (e.g., GradMatchPB-WARM) for both accuracy and training time. Is this difference mainly coming from the warm start? If so, would warm start help to improve the performance of the RBFNN Coreset? | Why is the training time with a warm start shorter than the one without (e.g., Table 1)? Shouldn't train on the whole dataset increase the training time?
>
> **Answer:** We thank the reviewer for the keen observation. We first explain why the time taken for training models with a warm start is almost the same as without. At (Killamsetty et al., 2021a), the WARM variant is defined as follows:
> Let $T$ be the total number of desired epochs, (in our experiments is set to be $300$).  The goal, in this case, is to train for $T_f$ epochs on the whole data, and then for $T_s$ epochs on the chosen subsets, while making sure that the overall time of training is equal to training $T$ epochs on a chosen subset. To do so, we set $T_s$ as a fraction $\kappa$ of the total number of epochs, i.e., $T_s:= \kappa T$ and $T_f:= \frac{T_s k}{n}$, where $k$ is the subset size, and $n$ is the size of the whole dataset (we use $\kappa = 0.5$).
>
> The time taken with the warm-start variant converges around the time that’s taken when running without such variants.
>
> As for accuracy, we note that doing full training for the first few epochs helps obtain good warm-start models, resulting in much better convergence. Setting $T_f$ to a large value yields results similar to the full training with early stopping (which we use as one of our baselines) since there are not enough data-selection techniques. Following this insightful comment, we have added warm start experiments with our methods, resulting in better results (see the updated cifar10 table in the paper, in the coming days we will add results also for cifar100). Thank you so much for pointing this out!
>
> ---------------------------------------------------------------------------------------------------------------------------------------------------------------------------------------
>
> **Comment 2:** The standard deviation of the empirical results is not reported - I'm not fully convinced the differences are statistically significant.
>
> **Answer:** Added to the appendix; see Tables 4,5 and 6. Note that while our standard deviation is larger than that of GradMatch for instance, however, in most of our results our reported mean accuracy is larger. Indeed our standard deviation is larger since we are applying a sampling technique for computing a generic coreset.
> In addition, the lower bound on the accuracy of our is slightly larger than that of our competing methods in most of our experiments.
>
> ---------------------------------------------------------------------------------------------------------------------------------------------------------------------------------------
>
> **Comment 3:** In definition 2 where the function $U(p)$ is defined, the $P$ should be $p$?
>
> **Answer:** Yes, the definition should be $U(p)=p(DV^T)^{-1}$, i.e., the $P$ should be $p$. We have fixed this typo, thanks for the careful reading!

---

### Author Response · Authors · 2022-11-18
**We are still ready to answer any questions at short notice - Thank you**

Dear Reviewers, ACs, and SACs,

We want to express our gratitude for the openly communicated review process as well as the reviewers who contributed greatly to its value and success by offering several useful ideas and comments - we truly appreciate that.

As the time for the discussion period is drawing to a close, we would want to make clear that we are available immediately to address any lingering issues and/or comments at short notice. We believe this is especially crucial because we haven't had a chance to interact with the reviewers on their remaining concerns (if any).

At the same time, we believe we were able to more than adequately address all the reviewers’ concerns and issues.

We note that we added more figures to the appendix, reapplying the function approximation experiment, and uploaded a clearer figure as the reviewers requested. In addition, we have put some of the results obtained for the ImageNet dataset showing that our approach can scale to other datasets and is not limited to CIFAR10/100 or MNIST. These results are present in the updated version of our paper here.

We are running more experiments on other models as well. These results will be ready to be incorporated into the camera-ready version of our paper.


Thank you and we look forward to hearing back,
The authors

---

### Decision · Program_Chairs · 2023-01-20

**Decision:**

Reject

**Justification For Why Not Higher Score:**

The submission is a combination of two ideas, which have some problems in motivation, and experimental results without significance.

**Justification For Why Not Lower Score:**

N/A

**Metareview: Summary, Strengths And Weaknesses:**

The submission proposes a coreset selection method based on (shallow) RBF neural networks, which are similar to RKHS coreset methods but with the theoretical ability to select RBF centers, an a number of RBF functions that can be set independently from the number of data points.  With this coreset selection procedure, the authors propose to use the selected points to train networks with less data, similar to the approaches proposed in Killamsetty et al. (2021a) (GRAD-MATCH), Mirzasoleiman et al. (2020a) (CRAIG), and in Killamsetty et al. (2021b) (GLISTER), which are the competing methods.  There is some new theoretical work on the construction of RBFNN coresets, resulting in theoretical results perhaps comparable to those available for RKHS coreset methods.  The submission received reviews around the borderline threshold.  There are several concerns about the paper in its current form: motivation and comparison to the current state of the art.

Motivation: the authors argue that a key benefit of the proposed approach is that there is one coreset for all subsequent networks, as the coreset method is independent of any subsequent network architecture (Section 5).  This is equally true for RKHS coresets, which have a rich literature and are from a function class with the same universal function approximation guarantee used to motivate the current method.  It seems that any coreset method could be used in the same setting, which indicates that additional experiments would be necessary to show why the RBFNN method, which is far from standard, is a better choice.

Experiments: Comparing Table 1 with the standard deviations in the appendix is different, and it is not clear that with 5 runs statistical significance can be established.  The authors' response in the rebuttal is difficult to parse, but does not claim significance and seems to acknowledge that it is not necessarily present: "Note that while our standard deviation is larger than that of GradMatch for instance, however, in most of our results our reported mean accuracy is larger. Indeed our standard deviation is larger since we are applying a sampling technique for computing a generic coreset. In addition, the lower bound on the accuracy of our is slightly larger than that of our competing methods in most of our experiments."  The values in 1, 2 & 3 have results in bold, when in some cases the values differ only in the second decimal place compared with competing methods.

On the balance, the submission explores an interesting question of generating coresets and using them in NN training.  There are essentially two main sets of contributions: 1) theoretical work developing a coreset theory for RBFNNs, and 2) empirical results showing their effect in training NNs with fewer data compared to several existing methods.  It is not entirely clear why RBFNNs would be selected over better studied RKHS coresets, and the empirical results are inconclusive compared with the chosen methods.

**Summary Of Ac-Reviewer Meeting:**

concerns regarding motivation for method and empirical results communicated